# Yeast cell responses and survival during periodic osmotic stress are controlled by glucose availability

**Fabien Duveau[1,2], Céline Cordier[3], Lionel Chiron[3], Matthias Le Bec[3], Sylvain Pouzet[3], Julie Séguin[1], Artémis Llamosi[1], Benoit Sorre[1,3], Jean-Marc Di Meglio[1], Pascal Hersen[1,3]\***

[1]Laboratoire Matière et Systèmes Complexes, UMR 7057 CNRS & Université Paris Diderot, 10 rue Alice Domon et Léonie Duquet, Paris, France; [2]Laboratoire de Biologie et Modélisation de la Cellule, Ecole Normale Supérieure de Lyon, CNRS, UMR 5239, Inserm, U1293, Université Claude Bernard Lyon 1, 46 allée d'Italie F-69364, Lyon, France; [3]Laboratoire Physico Chimie Curie, UMR168, Institut Curie, 16 rue Pierre et Marie Curie, 75005, Paris, France

**\*For correspondence:**
pascal.hersen@curie.fr

**Competing interest:** The authors declare that no competing interests exist.

**Abstract** Natural environments of living organisms are often dynamic and multifactorial, with multiple parameters fluctuating over time. To better understand how cells respond to dynamically interacting factors, we quantified the effects of dual fluctuations of osmotic stress and glucose deprivation on yeast cells using microfluidics and time-lapse microscopy. Strikingly, we observed that cell proliferation, survival, and signaling depend on the phasing of the two periodic stresses. Cells divided faster, survived longer, and showed decreased transcriptional response when fluctuations of hyperosmotic stress and glucose deprivation occurred in phase than when the two stresses occurred alternatively. Therefore, glucose availability regulates yeast responses to dynamic osmotic stress, showcasing the key role of metabolic fluctuations in cellular responses to dynamic stress. We also found that mutants with impaired osmotic stress response were better adapted to alternating stresses than wild-type cells, showing that genetic mechanisms of adaptation to a persistent stress factor can be detrimental under dynamically interacting conditions.

## eLife assessment

This study presents **important** findings on how cells sense and respond to their surroundings, in particular when two environmental signals are presented periodically, in alternation or conjunction. The **compelling** analyses reveal some unexpected behaviors that could not have been drawn, from simpler experimental designs, related to the dynamic interplay between the starvation and hyper-osmotic stress responses in budding yeast, exemplifying that applying complex signals can unveil new biological insights, even for well-studied systems. The work will be of broad interest to researchers interested in fungal biology, dynamic systems, cell signaling, and cell biology.

## Introduction

Cells have evolved to survive in a broad range of environmental conditions with multiple factors (e.g. temperature, nutrients, light, humidity, pathogens, etc.) varying in space and time. They can monitor their environment and constantly adapt their physiology to stress caused by environmental fluctuations. Experiments in which cells are dynamically probed with time-varying stress signals are required to obtain a quantitative understanding of how signaling pathways and gene regulatory networks confer

cellular adaptability to environmental changes (*Paliwal et al., 2008*). The development of microfluidics systems to study the frequency responses of cellular functions (*Bennett and Hasty, 2009*; *Kaiser et al., 2018*; *Hersen et al., 2008*; *Mettetal et al., 2008*) (e.g. signaling pathways, gene regulatory networks) has been instrumental in the adoption of the concepts of dynamic systems and information processing in biology. More recent methodological developments in the field of control theory have enabled time-varying perturbations to be used to control cellular gene expression or signaling pathways via computer-based external feedback loops (*Harrigan et al., 2018*; *Milias-Argeitis et al., 2016*; *Lugagne et al., 2017*; *Uhlendorf et al., 2012*; *Rullan et al., 2018*; *Banderas et al., 2020*). In short, methods are now mature to study cells as dynamical systems.

Most studies of cellular stress responses have focused on a single environmental stress in an otherwise maintained environment. However, how cells respond to stress often depends on the interaction between several environmental factors. For instance, changing the metabolic environment (e.g. carbon source type and concentration) can profoundly affect cell physiology (e.g. respiration and fermentation in yeast) and alter stress responses (*Babazadeh et al., 2017*; *Rodaki et al., 2009*). More generally, resource allocation (i.e. how cellular resources are shared between several cellular functions) is an important fundamental (*Weiße et al., 2015*; *Metzl-Raz et al., 2017*) (e.g. understanding growth laws) and applied topic (*Ceroni et al., 2018*) (e.g. design of robust synthetic gene circuits and bioproduction). Specifically, cells face decision-making problems when exposed to stress and to variation in their metabolic environment. Routing resources toward stress response mechanisms may deprive other important processes (e.g. cellular maintenance, proliferation) and decrease competitive fitness in an environment periodically scarce of metabolic resources. Conversely, routing resources toward cell proliferation may reduce survival in stressful conditions and therefore also reduce fitness. The trade-off between proliferation and stress responses can be an important determinant of cell fitness in a dynamic environment (*Weiße et al., 2015*; *Metzl-Raz et al., 2017*; *Granados et al., 2017*; *Reimers et al., 2017*). Yet, the extent to which what is known in rich, constant metabolic conditions remains valid under low or fluctuating nutrient availability remains an open question. Here, we address this broad question by studying the synergistic and antagonistic effects of time-varying osmotic stress and glucose deprivation on the growth of budding yeast cells.

The adaptation to hyperosmotic stress in the budding yeast *Saccharomyces cerevisiae* involves an adaptive pathway—the HOG pathway—that has been extensively described at the molecular level (*de Nadal and Posas, 2022*) as well as biophysical and integrative levels through mathematical and computational models (*Klipp et al., 2005*; *Muzzey et al., 2009*; *Petelenz-Kurdziel et al., 2013*; *Sharifian et al., 2015*; *Krantz et al., 2009*; *Zi et al., 2010*; *Schaber et al., 2012*). Quantitative descriptions of the dynamics of osmotic stress responses were achieved using microfluidics to generate time-varying perturbation of the osmolarity of the environment while observing signaling activity and the transcriptional responses of key players in the HOG pathway at the single-cell resolution via time-lapse microscopy (*Hersen et al., 2008*; *Mettetal et al., 2008*; *Mitchell et al., 2015*).

When external osmolarity increases, accumulation of intracellular glycerol is required to restore the cellular osmotic balance (*Hohmann, 2002*). At the molecular level, osmotic stress signaling is orchestrated by a mitogen-activated protein kinase (MAPK) cascade, which culminates in double phosphorylation and nuclear accumulation of the MAPK protein Hog1p and differential regulation of hundreds of genes (*Saito and Posas, 2012*; *Gasch et al., 2000*). In particular, GPD1 (NAD-dependent glycerol-3-phosphate dehydrogenase), a key enzyme involved in the production of glycerol from glucose, is upregulated after hyperosmotic stress (*Figure 1a*). Phosphorylated Hog1p also triggers several processes in the cytoplasm that are essential for osmoregulation (*Muzzey et al., 2009*; *Petelenz-Kurdziel et al., 2013*), including cell cycle arrest (*Escoté et al., 2004*; *Clotet et al., 2006*; *Duch et al., 2013*). Dynamically, the HOG pathway behaves as a low-pass filter that drives (perfect) adaptation through at least two layers of feedback loops that allow for deactivation of the pathway (*Muzzey et al., 2009*) (transcriptionally and within the cytoplasm). Notably, the HOG pathway can be hyperactivated when stimulated at high frequencies, which drastically slows down the cell cycle (*Mitchell et al., 2015*). Although very informative—and an excellent example of how biological and physical concepts can be combined to obtain a comprehensive description of gene regulatory network dynamics—these studies were carried out in a glucose-rich environment, which insulates metabolic needs from osmotic stress adaptation requirements. Glucose is not only needed for growth, but also for production of glycerol and the transcriptional feedback loop that deactivates the HOG pathway

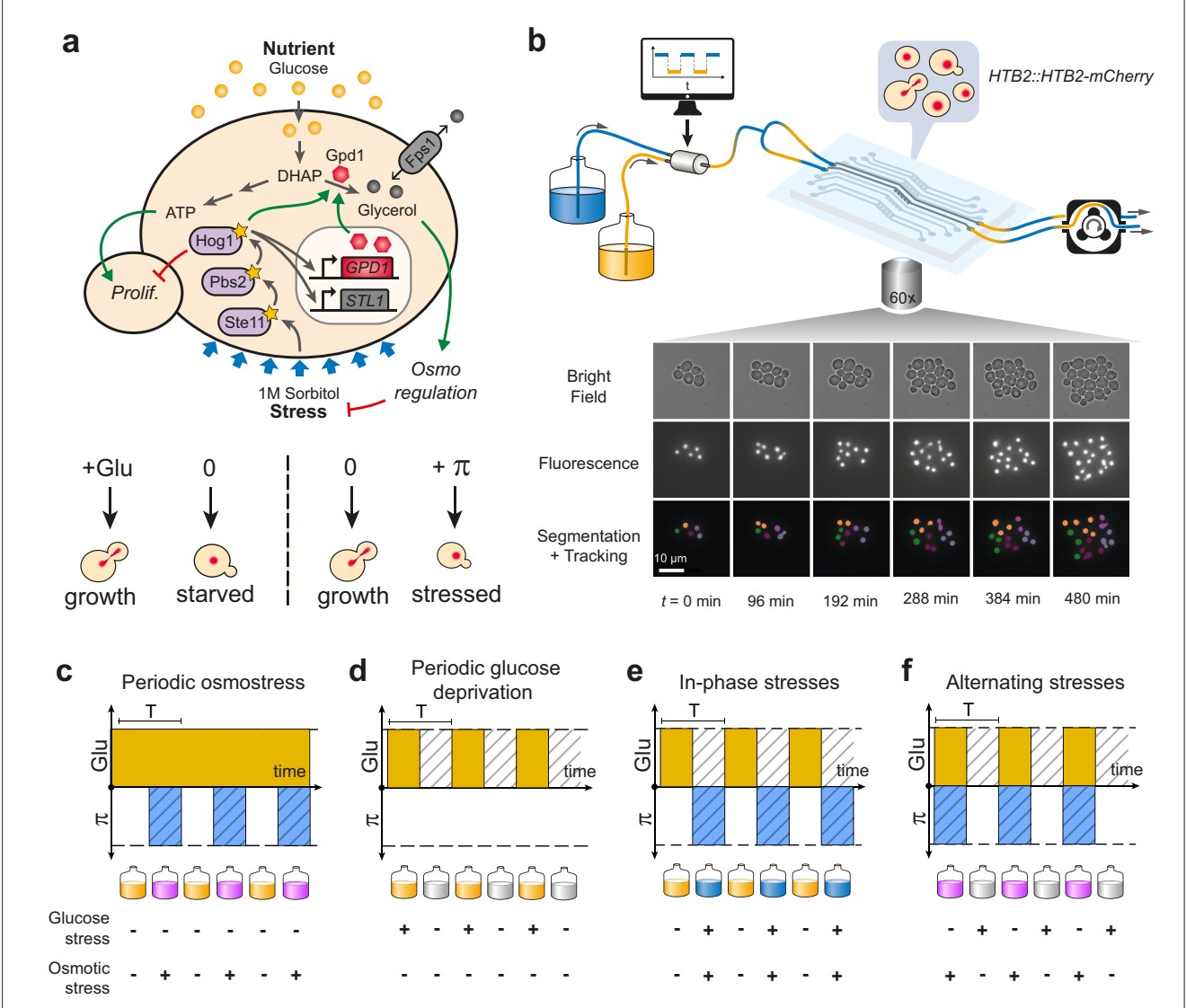

**Figure 1.** Live imaging of yeast cells grown in periodically fluctuating environments. (**a**) Overview of the hyperosmotic stress response in yeast. Both glucose deprivation and osmotic stress lead to cell cycle arrest—through different molecular mechanisms. Yeast cells maintain osmotic equilibrium by regulating the intracellular concentration of glycerol. Glycerol synthesis is regulated by the activity of the HOG MAP kinase cascade that acts both in the cytoplasm (fast response) and on the transcription of target genes in the nucleus (long-term response). For simplicity, we only represented on the figure genes and proteins involved in this study. (**b**) Sketch of the microfluidic setup used to generate a time-varying environment and achieve time-lapse imaging of yeast cells. Bright-field and fluorescence images are captured every 6 min at 25 positions for 12–24 hr depending on the experiment. Nuclei expressing HTB2-mCherry fusion protein are segmented and tracked over time to compute the cell division rate as a function of time. Scale bar represents 10 µm. (**c–f**) The four periodically varying environments used in this study. (**c**) Periodic osmotic stress: cells are periodically exposed to hyperosmotic stress (1 M sorbitol) in a constant glucose environment (2% or 0.1%). (**d**) Periodic glucose deprivation: environment alternates between presence and absence of glucose. (**e**) In-phase stresses (IPS): periodic exposure to glucose in absence of hyperosmotic stress followed by glucose depletion with hyperosmotic stress (1 M sorbitol). (**f**) Alternating stress (AS): periodic exposure to glucose with hyperosmotic stress (1 M sorbitol), followed by glucose depletion without hyperosmotic stress. (**c–f**). Hatching represents stress; blue indicates presence of sorbitol; orange, presence of glucose.

The online version of this article includes the following source data and figure supplement(s) for figure 1:

**Figure supplement 1.** Dynamics of media switching inside the microfluidic chip.

**Figure supplement 1—source data 1.** Source data for **Figure 1—figure supplement 1**.

(*Babazadeh et al., 2017*; *Muzzey et al., 2009*); thus, cells may employ a decision mechanism to share glucose internally between these processes, particularly when glucose is scarce, or its availability fluctuates. More generally, despite the known importance of the metabolic state in cellular adaptation to stress, the systemic interactions between cellular maintenance, growth, and stress responses remain unexplored.

We address this question by monitoring the growth of yeast cells subjected to periodic variations in both osmolarity and glucose availability. To determine how resource allocation impacted cell growth, we compared two regimes of dual fluctuations that differed in the phasing of hyperosmotic stress and glucose deprivation. We showed that cell division rates, death rates, and biological responses at the signaling and transcription levels are different when cells are exposed simultaneously (in-phase stresses [IPS]) or alternatively (alternating stresses [AS]) to glucose deprivation and hyperosmotic stress. Therefore, yeast responses to osmotic stress are regulated by the presence of external glucose, indicating that the metabolic environment is a key factor when quantitatively assessing stress response dynamics. More globally, our study suggests that applying dual periodic perturbations is a powerful method to probe cellular dynamics at the system level and, more specifically, to clarify the role of the metabolic environment in the dynamics of cellular decision-making.

## Results

### A microfluidic system to study the interaction between two environmental dynamics

We used a custom microfluidic device to monitor the growth of yeast cells exposed to periodic environmental fluctuations for up to 24 hr. Cells were imaged every 6 min in microfluidic chips containing five independent sets of channels connected to five growth chambers (*Figure 1b*; *Figure 1—figure supplement 1*), allowing five different conditions per experimental run, with five technical replicates for each condition. Computer-controlled fluidic valves were programmed to generate temporal fluctuations of the media dispensed to cells with rapid transitions (<2 min) from one medium to another (*Figure 1—figure supplement 1*). The rate of cell division was then quantified using automated image analyses (see Materials and methods). With this experimental system it is possible to determine not only how temporal fluctuations of individual parameters of the environment (e.g. a repeated stress or carbon source fluctuations) impact cell proliferation but also what are the impacts of the dynamic interactions of two environmental parameters. Here, we specifically study how periodic fluctuations of a metabolic resource (glucose concentration switching between 0% and either 2% or 0.1% wt/vol) and osmotic stress (sorbitol concentration switching between 0 and 1 M) interact to alter the proliferation of yeast cells (*Figure 1*).

### Division rate correlates negatively with the frequency of osmotic stress but positively with the frequency of glucose availability

To determine how the temporal dynamics of osmotic stress altered cell proliferation, we first measured the division rate of yeast cells exposed to fluctuations between 1 M sorbitol and no sorbitol at periods ranging from 12 to 480 min. In these experiments the time-averaged osmotic concentration was constant (i.e. cells were exposed to 1 M sorbitol half of the time in all conditions), which is important when studying the effects of the *frequency*, and not *intensity*, of osmotic stress on cell dynamics. The average division rate strongly decreased as the frequency of osmotic shock increased (*Figure 2a*), both in 2% glucose (2.2-fold reduction of division rate between periods of fluctuation $T$=192 min and $T$=12 min) and in 0.1% glucose (3.8-fold reduction of division rate between periods of fluctuation $T$=192 min and $T$=12 min). These results are consistent with findings from *Mitchell et al., 2015*, who attributed the drastic decrease in cellular growth observed at high frequency of osmotic shocks to overactivation of the HOG pathway. However, we also observed a clear negative relationship between the frequency of hyperosmotic stress and the division rate of HOG pathway mutants (*Figure 2—figure supplement 3*), indicating that the growth slowdown was not only explained by overactivation of the HOG pathway. The temporary reduction of division rate observed in response to a hyperosmotic shock in wild-type (*Figure 2e–f*; *Figure 2—figure supplement 1*; *Figure 2—figure supplement 2a-c*) and mutant (*Figure 2—figure supplement 3*) cells could also contribute to the negative relationship between division rate and frequency of osmotic stress. This negative relationship and the fact that

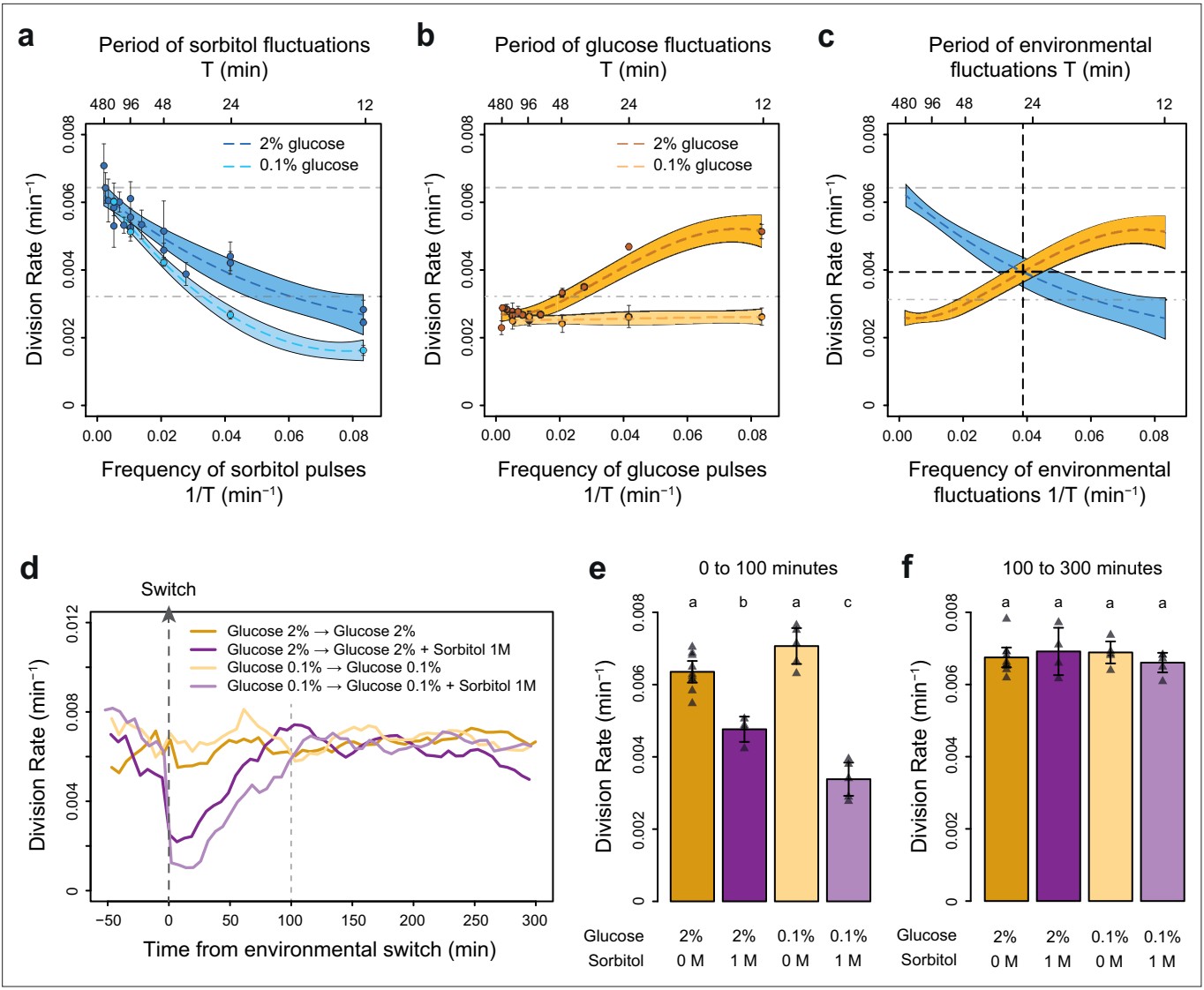

**Figure 2.** The frequencies of osmostress and glucose availability impact cell division rates in opposite ways. (**a, b**) Impact of the frequency of periodic osmotic stress (**a**) and glucose deprivation (**b**) on the average division rate. Each dot shows the mean division rate measured in 2 to 5 different growth chambers of the microfluidic chip. Error bars are 95% confidence intervals of the mean. Colored dotted lines are Loess regressions obtained using a smoothing parameter of 0.66. Colored areas represent 95% confidence intervals of the regression estimates. Gray dashed lines show the average division rate in the absence of sorbitol (no osmotic stress) in 2% glucose (top line) and dash-dotted lines show half this average division rate (bottom line). (**c**) Overlay of the Loess regressions shown in (**a**) and (**b**) at 2% glucose. The frequency and division rate at which the two regression curves intersect are highlighted by vertical and horizontal black dotted lines. (**d**) Temporal dynamics of the division rate of cells exposed to sustained hyperosmotic stress (purple lines, 1 M sorbitol added at *t*=0 min) or to standard conditions (orange lines) with 2% glucose (darker lines) or 0.1% glucose (lighter lines). Each curve represents the 'instantaneous' division rate calculated every 6 min across sliding windows of 36 min (see Materials and methods) and averaged for cells imaged at 2 to 10 positions in the microfluidic chip. (**e, f**) Cell division rates measured (**e**) from 0 to 100 min and (**f**) from 100 to 300 min after the addition of 1 M sorbitol. Triangles represent the average division rate measured in different growth chambers of the microfluidic chips. Error bars are 95% confidence intervals of the mean division rate among growth chambers. The initial number of cells analyzed among replicates ranged from 75 to 372 in (**a**), from 58 to 342 in (**b**), and from 220 to 776 in (**d–f**).

The online version of this article includes the following source data and figure supplement(s) for figure 2:

**Source data 1.** Source data for *Figure 2*.

**Figure supplement 1.** Impact of a single hyperosmotic shock on cell division rate.

**Figure supplement 1—source data 1.** Source data for *Figure 2—figure supplement 1*.

**Figure supplement 2.** Temporal dynamics of division rate under periodic osmotic stress and under periodic glucose depletion.

**Figure supplement 2—source data 1.** Source data for *Figure 2—figure supplement 2*.

*Figure 2 continued on next page*

*Figure 2 continued*

**Figure supplement 3.** Division rates of HOG pathway mutants under constant and periodic osmotic stress.

**Figure supplement 3—source data 1.** Source data for *Figure 2—figure supplement 3*.

cellular growth can rapidly recover after exposure to high osmolarity (*Figure 2e–f*) both indicate that yeast cells are more sensitive to *repeated* than *persistent* hyperosmotic stress.

Next, we wondered whether the frequency of a different type of environmental fluctuation would also affect cell division rate. To answer this question, we quantified the division rate of cells exposed to periodic transitions between a medium without carbon source and the same medium complemented with either 0.1% or 2% glucose at periods ranging from 12 to 480 min. In contrast to the negative effect of osmotic stress frequency, we observed a positive relationship between the frequency of glucose availability and division rate (*Figure 2b*): cells divided faster when glucose availability fluctuated rapidly (0.0051 division/min at a fluctuation period $T$=12 min, corresponding to a doubling time of 136 min) than slowly (0.0027 division/min at a fluctuation period $T$=192 min, corresponding to a doubling time of 257 min). However, this behavior was only observed in 2% glucose: the *frequency* of glucose availability did not significantly impact the division rate in 0.1% glucose (*Figure 2b*). Under periodic fluctuations of 2% glucose, the division rate was lower during half-periods without glucose than during half-periods with glucose (*Figure 2—figure supplement 2*), as expected. However, this difference depended on the frequency of glucose fluctuations: the average division rate during half-periods without glucose was higher at high frequency (small period) than at low frequency (large period) of fluctuations (*Figure 2—figure supplement 2*). Therefore, the effect of the *frequency* of glucose availability on the division rate in 2% glucose is likely due to a delay between glucose removal and growth arrest: cell proliferation never stops when the frequency of glucose depletion is too fast.

Overall, we observed two opposing patterns of cell proliferation when we varied the temporal dynamics of the metabolic environment and external osmolarity. The division rates were highest for low-frequency sorbitol fluctuations (0.0064 division/min at a fluctuation period $T$=384 min) and high-frequency 2% glucose fluctuations (0.0051 division/min at a fluctuation period $T$=12 min); both of these values are close to the division rate in constant 2% glucose (0.0066 division/min). Therefore, with respect to their division rate, cells behave as a low-pass filter for osmotic stress but as a high-pass filter for glucose fluctuations. Moreover, the division rate is similar when the frequencies of glucose availability and sorbitol exposure are both equal to 0.039 min$^{-1}$ (intersection of the two curves on *Figure 2c*), corresponding to a period of 26 min and a division rate of 0.004 division/min. Since current models of the hyperosmotic stress response do not consider interactions with glucose metabolism, whether simultaneous fluctuations of glucose availability and osmotic stress affect cell growth additively or synergistically remains an open question. More generally, characterizing how cells respond to the dynamic phasing of two environmental components is fundamental for understanding how a living system can adapt to complex environmental changes. For these reasons, we next used our microfluidic system to quantify the division rate of cells exposed to *dual* periodic fluctuations of glucose availability and osmotic stress.

## Division rate depends on the phasing of the two stresses

To determine whether glucose availability during hyperosmotic stress impacted cell growth in dynamic conditions, we compared cell division rates under two regimes of dual periodic fluctuations that only differed in the phasing of glucose and sorbitol fluctuations. In the 'IPS' regime, glucose depletion and 1 M sorbitol stresses were applied simultaneously for half a period followed by the addition of 2% (or 0.1%) glucose and the removal of sorbitol for the other half of each period of fluctuations (*Figure 1e*). In the 'AS' regime, glucose depletion and 1 M sorbitol were applied alternatively for half a period each (*Figure 1f*). We first subjected cells to dual fluctuations at a period of 24 min with 2% glucose, corresponding approximately to the period at which the division rate was the same when we only varied glucose availability *or* osmolarity (*Figure 2c*). Under both IPS and AS conditions, the division rate was more than twofold lower than under periodic fluctuations of only glucose or sorbitol (*Figure 3a*), showing that dual environmental fluctuations have a non-additive, synergistic impact on cell growth. Strikingly, cells divided about twice as fast under IPS condition (1.67×10$^{-3}$ division/min, corresponding to an average doubling time of 415 min) than under AS condition (9.4×10$^{-4}$ division/min, corresponding to an average doubling time of 737 min) when the fluctuation period was 24 min

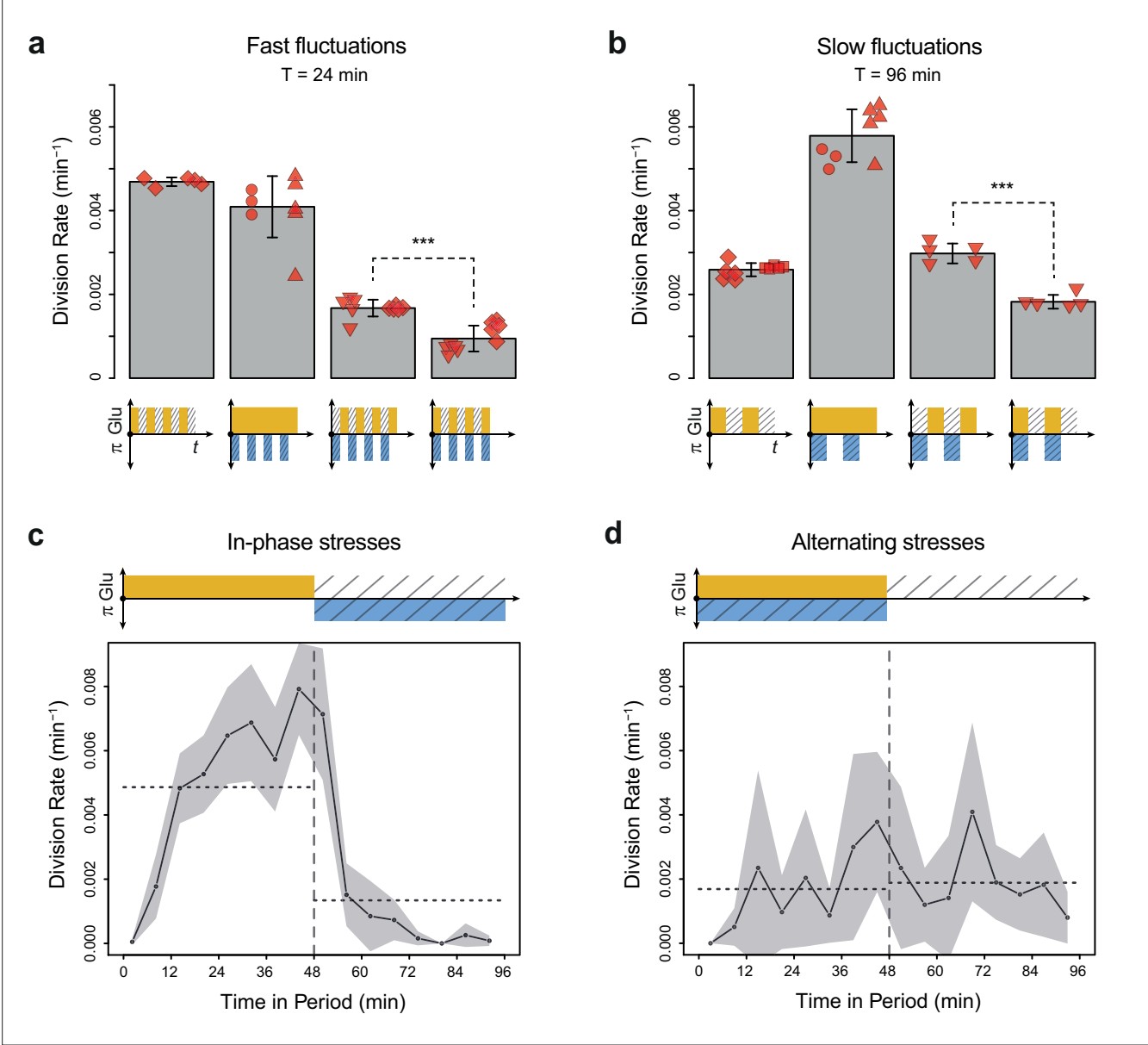

**Figure 3.** Cell division rate depends on the phasing of hyperosmotic stress and glucose availability. (**a, b**) Division rates measured in four fluctuating conditions with a period of 24 min and a glucose concentration of 2% (20 g/L). The four conditions are periodic glucose deprivation, periodic osmostress, in-phase stresses (IPS), and alternating stresses (AS). Bars represent mean division rates among different growth chambers. Error bars are 95% confidence intervals of the mean. Red symbols show the average division rate in each growth chamber, with different symbols representing experiments performed on different days with different microfluidic chips. Mean division rates were compared between IPS and AS conditions using *t*-tests (***p<0.001). (**c, d**) Temporal dynamics of division rate during a period of 96 min in (**c**) IPS and (**d**) AS conditions for wild-type cells. Each dot shows the average division rate during a 6 min window centered on that dot for all fields of view sharing the same condition and for all periods in the experiment (average of 16 to 30 measurements per dot). Gray areas are 95% confidence intervals of the mean division rate. Horizontal dotted lines show the mean division rate for all data collected in each half-period. The colored bars represent the periodic fluctuations in glucose (orange) and/or sorbitol (blue); hatching represents stress. (**a–d**) Cells were grown under fluctuations of 2% glucose and 1 M sorbitol. The initial number of cells analyzed among replicates ranged from 124 to 318 in (**a**) and from 94 to 342 in (**b–d**).

The online version of this article includes the following source data and figure supplement(s) for figure 3:

**Source data 1.** Source data for *Figure 3*.

**Figure supplement 1.** Division rates under regimes of in-phase stresses (IPS) and alternating stresses (AS) with different fluctuation periods and glucose concentrations.

**Figure supplement 1—source data 1.** Source data for *Figure 3—figure supplement 1*.

(*t*-test, p=1.35 × 10⁻⁵; *Figure 3a*, *Figure 3—figure supplement 1a and b*) or 96 min (2.98×10⁻³ division/min in IPS vs 1.83×10⁻³ division/min in AS; p=4.10 × 10⁻⁵; *Figure 3b*). A similar pattern of faster growth was observed under IPS and AS conditions when we used 0.1% glucose instead of 2% glucose, for both a fluctuation period of 24 min (0.84×10⁻³ division/min under IPS vs 0.54×10⁻³ division/min under AS; *t*-test, p=8.03 × 10⁻⁵; *Figure 3—figure supplement 1c*) and 96 min (2.24×10⁻³ division/min under IPS vs 1.17×10⁻³ division/min under AS; *t*-test, p=6.80 × 10⁻³; *Figure 3—figure supplement 1d*). Cells also displayed strikingly different temporal dynamics of division rates under IPS and AS conditions (*Figure 3c and d*). Under IPS condition, the division rate fluctuated largely over time: after the transition to 2% glucose, the division rate quickly increased to reach a plateau (4.86×10⁻³ division/min on average during the half-period with 2% glucose); after the transition to 1 M sorbitol in the absence of glucose, cell division was greatly slowed down (1.34×10⁻³ division/min during the half-period without glucose). In contrast, the division rate remained much more constant over time under AS condition: the average division rate was 1.69×10⁻³ division/min during the half-period with 2% glucose and 1 M sorbitol and 1.89×10⁻³ division/min during the half-period without glucose and sorbitol. Therefore, cells appear to use glucose more efficiently for growth under IPS than AS conditions. Collectively, these results further demonstrate that the timing of both glucose availability and osmotic stress matters: cells grow more slowly when facing periodic alternation of the two stresses (AS) than when facing periodic co-occurrence of these stresses (IPS).

The slower cell division rate observed under AS when compared to IPS could be explained by the allocation of intracellular glucose to the osmotic stress response under AS when cells are exposed to glucose and sorbitol simultaneously, leaving less glucose available for growth. Indeed, in response to hyperosmotic stress glycerol is synthesized from a glycolysis intermediate (DHAP) derived from glucose (*Norbeck et al., 1996*). Under this hypothesis, glucose would only be fully allocated to growth in the absence of hyperosmotic stress, which occurred under IPS but not AS.

## Slowdown of cell proliferation under AS is independent of HOG pathway activity

To test the hypothesis that the allocation of glucose toward glycerol synthesis explained the slower division rate observed under AS relative to IPS, we compared the division rate of mutants with impaired glycerol regulation under IPS and AS conditions. These mutant strains carried deletions of *HOG1* (HOG pathway MAPK), *PBS2* (MAPKK upstream of Hog1p), *STE11* (MAPKKK upstream of Pbs2p), *FPS1* (aquaglyceroporin regulated by Hog1p), *GPD1* (glycerol-3-phosphate dehydrogenase regulated transcriptionally and post-transcriptionally by the HOG pathway) or *GPD2* (paralog of *GPD1*). As expected, these mutants showed no growth defect in the absence of hyperosmotic stress and most mutants showed decreased division rates when exposed to constant hyperosmotic stress (*Figure 2—figure supplement 3*). At a fluctuation period of 24 min, the division rate was significantly lower under AS than IPS for almost all mutants (*hog1Δ*, *pbs2Δ*, *gpd1Δ*, *gpd2Δ*, *gpd1*; *gpd2Δ* and *fps1Δ*) both with fluctuations of 2% glucose (*Figure 4a*) and 0.1% glucose (*Figure 4—figure supplement 1a*). However, the *ste11Δ* mutant exhibited similar division rates under AS and IPS (*Figure 4a*). At a fluctuation period of 96 min, the division rates of the two mutants we tested, *ste11Δ* and *pbs2Δ*, were significantly lower under AS than IPS (*Figure 4—figure supplement 1b and c*). In addition, the temporal dynamics of division rates were similar for the wild-type strain, *pbs2Δ* mutant (*Figure 4b and c*) and *ste11Δ* mutant (*Figure 4—figure supplement 1d and e*). In conclusion, mutations known to reduce intracellular accumulation of glycerol did not attenuate the growth differences that we observed in the wild-type strain between IPS and AS conditions. Therefore, allocation of glucose toward glycerol synthesis during hyperosmotic stress is not responsible for the lower division rate observed under AS than IPS.

## No evidence for a specific role of glucose starvation, glycogen storage, or stress-induced arrest of the cell cycle in the reduced division rate observed during AS

We next tested alternative hypotheses to understand why cells grew slower under AS than IPS condition. Glucose starvation was previously shown to induce fast inhibition of transcription (*Jona et al., 2000*) and translation initiation (*Ashe et al., 2000*; *Joyner et al., 2016*), leading to cell growth reduction. This phenomenon may explain the slower division rate observed in AS condition than in IPS

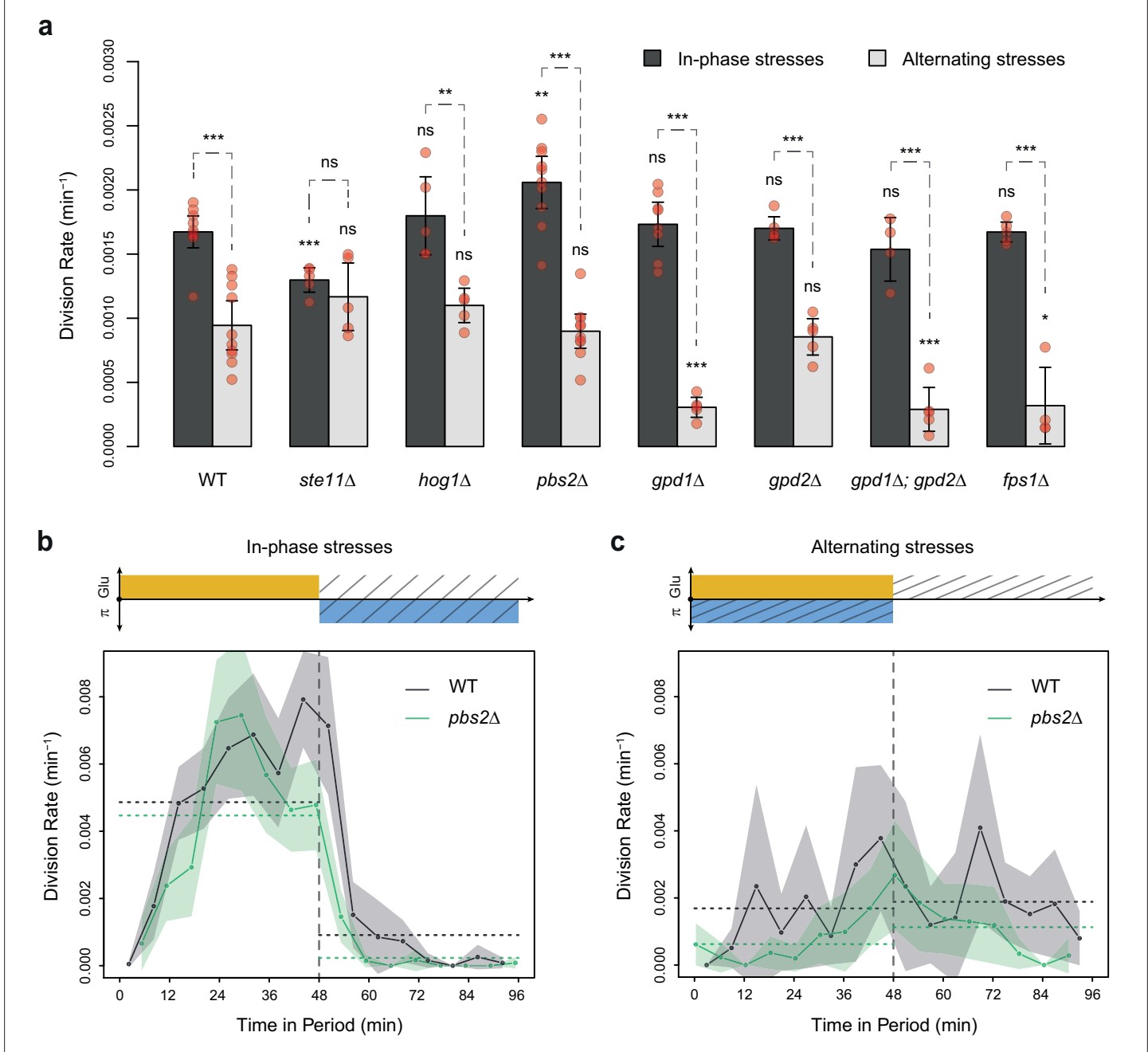

**Figure 4.** HOG pathway mutants grow faster under in-phase stresses (IPS) than under alternating stresses (AS). (**a**) Division rates measured during growth in IPS and AS conditions with 2% glucose and a fluctuation period of 24 min. Bars show the mean division rate measured in different growth chambers of the microfluidic chip. Error bars are 95% confidence intervals of the mean. Red symbols show the average division rate in each growth chamber. Results of *t*-tests comparing the wild-type and mutant strains under the same condition are indicated above each bar; results comparing the same strain under different conditions are shown above each pair of bars (ns: p>0.05; *0.01<p<0.05; **0.001<p<0.01; ***p<0.001). (**b, c**) Temporal dynamics of division rate during a period of 96 min in (**b**) IPS and (**c**) AS conditions for wild-type (black) and *pbs2Δ* mutant (green) cells. Each dot shows the division rate during a 6 min window centered on that dot and averaged for all fields of view sharing the same condition and all periods in the experiment (average of 16 to 30 measurements per dot). Gray and green areas are 95% confidence intervals of the mean division rate. Horizontal dotted lines show the mean division rate for all data collected in each half-period. The colored bars represent the periodic fluctuations in glucose (orange) and/ or sorbitol (blue); hatching represents stress. The initial number of cells analyzed among replicates ranged from 97 to 467 in (**a**) and from 124 to 145 in (**b, c**).

The online version of this article includes the following source data and figure supplement(s) for figure 4:

**Source data 1.** Source data for *Figure 4*.

*Figure 4 continued on next page*

*Figure 4 continued*

**Figure supplement 1.** Division rates of HOG pathway mutants during in-phase stresses (IPS) and alternating stresses (AS) with different fluctuation periods and glucose concentrations.

**Figure supplement 1—source data 1.** Source data for *Figure 4—figure supplement 1*.

**Figure supplement 2.** Comparing division rates during in-phase stresses and alternating stresses of wild-type cells exposed to periodic glucose (gray) or periodic galactose (green) and of mutant cells with impaired glycogen accumulation (*glc3* deletion) or impaired cell cycle arrest in response to hyperosmotic stress (*sic1* alleles).

**Figure supplement 2—source data 1.** Source data for *Figure 4—figure supplement 2*.

condition, because rapid arrest of the cell cycle after glucose starvation could have smaller impact on global division rate when occurring concurrently (IPS) rather than alternatively (AS) with hyperosmotic stress that also leads to fast growth reduction. We tested this hypothesis by growing wild-type cells under IPS and AS conditions with galactose instead of glucose as a carbon source, because transcriptional and translational inhibition was not observed after galactose starvation in previous studies (*Ashe et al., 2000*; *Jona et al., 2000*). We observed a significant reduction of division rate under AS condition relative to IPS condition when using galactose as a carbon source, similar to what we observed in glucose (*Figure 4—figure supplement 2*). Therefore, fast inhibition of transcription and translation occurring after glucose starvation but not after galactose starvation does not contribute significantly to the slower growth in AS condition.

Yeast cells accumulate carbohydrate reserves such as glycogen to cope with nutrient starvation (*Wilson et al., 2010*; *François and Parrou, 2001*), and this reserve of glycogen can be mobilized during hyperosmotic stress (*Bonny et al., 2021*; *Parrou et al., 1997*). Glycogen may accumulate less under AS condition because glucose is only available during hyperosmotic stress, leading to slower growth. To test the hypothesis that glycogen storage may contribute to the difference of division rates observed between AS and IPS conditions, we quantified the division rates of *glc3Δ* mutant cells with impaired glycogen synthesis in these two conditions. Once again, we observed a significantly lower division rate of *glc3Δ* cells in AS condition relative to IPS condition similar to what was observed in wild-type cells, suggesting that glycogen storage was not significantly contributing to this difference.

Third, we tested the impact of point mutations in the cyclin inhibitor Sic1p on division rates in AS and IPS conditions. In response to hyperosmotic shock, residue 173 of Sic1p is phosphorylated by Hog1p, resulting in Sic1p stabilization and cell cycle arrest at the G1 phase (*Escoté et al., 2004*). Since hyperosmotic stress and glucose starvation both lead to cell growth arrest, cell division is expected to halt twice more frequently when hyperosmotic stress and glucose starvation are applied alternatively than when they are applied simultaneously, which could lead to the difference of division rates observed between AS and IPS conditions in a way that depends on Sic1p regulation. However, *sic1(T173A)* mutant cells (unphosphorylatable Sic1p) and *sic1(T172E)* mutant cells (constitutive Sic1p stabilization) showed a similar decrease of division rate in AS condition relative to IPS condition as observed in wild-type cells (*Figure 4—figure supplement 2*). The mechanism(s) responsible for the lower division rate in AS condition relative to IPS condition therefore remain(s) elusive.

## Cell death depends on the dynamics of the two stresses

We noticed a high proportion of wild-type cells dying under AS (*Figure 5a*): some cells suddenly burst with their nucleus staying in the growth chamber, others became opaque and stopped growing with their nucleus remaining completely still (the nucleus of living cells wobbled over time). These death events mostly occurred within minutes of the transition from medium containing 2% glucose and 1 M sorbitol to medium without glucose and sorbitol (*Figure 5—figure supplement 1d*), suggesting cell lysis occurred due to hypo-osmotic shock following removal of 1 M sorbitol. Cell death was less frequent under IPS than AS conditions for the wild-type strain (*Figure 5a–c*), even though the frequency of hypo-osmotic shock was the same in the two conditions (*Figure 5—figure supplement 1c and d*). We reasoned this could be due to lower intracellular accumulation of glycerol under IPS, when hyperosmotic stress is applied in the absence of glucose. Under AS, the presence of glucose during hyperosmotic stress could lead to faster intracellular accumulation of glycerol, resulting in stronger hypo-osmotic shock and cell lysis when the sorbitol concentration suddenly drops. Several pieces of evidence support this hypothesis. First, the rate of cell death should be reduced in mutants

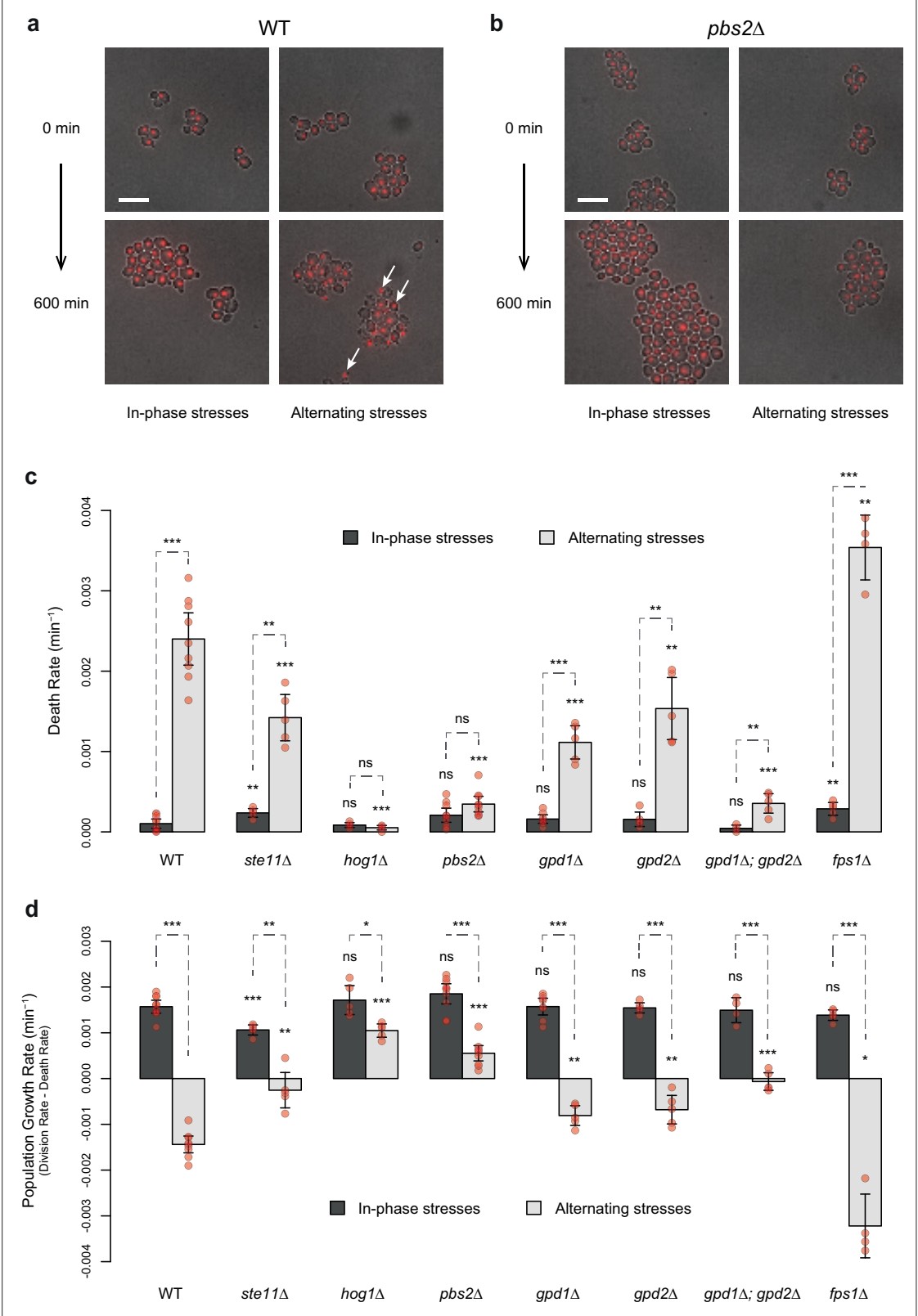

**Figure 5.** HOG pathway mutants exhibit a lower death rate and an increased population growth rate under in-phase stresses (IPS). (**a, b**) Images of wild-type (**a**) and *pbs2Δ* mutant (**b**) cells before (*t*=0 min) and after (*t*=600 min) growth in IPS (left) and alternating stress (AS) (right) conditions. Fluorescence and bright-field images were merged to visualize nuclei marked with HTB2-mCherry. White arrows indicate nuclei of representative dead cells. Scale bars represent 10 μm. (**c, d**) Death rates (**c**) and population growth rates (**d**) of the reference strain and seven deletion mutants under IPS

*Figure 5 continued on next page*

*Figure 5 continued*

and AS conditions. Population growth rates were calculated as the difference between division rates (*Figure 4b*) and death rates (**c**). Bars show mean rates measured in different growth chambers. Error bars are 95% confidence intervals of the mean. Red symbols show the average rate for each field of view. Results of *t*-tests comparing the wild-type and mutant strains under the same conditions are indicated above each bar; results comparing the same strain under different conditions are shown above each pair of bars (ns p>0.05; *0.01<p<0.05; **0.001<p<0.01; ***p<0.001). The initial number of cells analyzed among replicates ranged from 97 to 467 in (**c, d**). (**a–d**) Cells were grown under fluctuations of 2% glucose and 1 M sorbitol at a period of 24 min.

The online version of this article includes the following source data and figure supplement(s) for figure 5:

**Source data 1.** Source data for *Figure 5*.

**Figure supplement 1.** Dynamics of cell death and population growth rates under combined stresses and alternating stresses (AS).

**Figure supplement 1—source data 1.** Source data for *Figure 5—figure supplement 1*.

**Figure supplement 2.** Death rates of wild-type and *sic1* mutant cells under periodic fluctuations of hyperosmotic stress with or without fluctuations of glucose availability.

**Figure supplement 2—source data 1.** Source data for *Figure 5—figure supplement 2*.

with lower glycerol synthesis. Indeed, we observed significantly lower rates of cell death for all mutants tested (*ste11Δ*, *hog1Δ*, *pbs2Δ*, *gpd1Δ*, *gpd2Δ* and *gpd1Δ; gpd2Δ*) relative to the wild-type strain under AS, but not under IPS for which glycerol synthesis is frustrated even in wild-type cells due to the absence of glucose during hyperosmotic stress (*Figure 5c*). In particular, the death rate decreased from $2.4 \times 10^{-3}$ min$^{-1}$ for wild-type to $5.2 \times 10^{-5}$, $3.4 \times 10^{-4}$, and $3.5 \times 10^{-4}$ min$^{-1}$, respectively, for the *hog1Δ*, *pbs2Δ*, and *gpd1Δ; gpd2Δ* mutants under AS when the fluctuation period was 24 min. A similar pattern was observed for the *pbs2Δ* mutant when the fluctuation period was 96 min (*Figure 5—figure supplement 1a*), although the reduction in the death rate was less pronounced than for the period of 24 min. Conversely, we observed a higher death rate for the *fps1Δ* mutant ($3.5 \times 10^{-3}$ min$^{-1}$) under AS (*Figure 5c*), which is consistent with higher intracellular accumulation of glycerol in this mutant lacking the Fps1 aquaglyceroporin channel involved in glycerol export. Over a 10 hr AS experiment with a fluctuation period of 24 min, the death rate was lowest at the beginning of the experiment and was maximal during the last 5 hr of the experiment (*Figure 5—figure supplement 1e*). However, when the AS fluctuation period was 96 min, the maximum death rate occurred earlier (and stopped increasing after the second period) and the dynamics of cell death remained constant over multiple periods of osmotic fluctuation (*Figure 5—figure supplement 1f*). Again, these observations are consistent with cell death being due to glycerol accumulation, since it takes time for cells to accumulate an amount of glycerol sufficient to cause bursting after hypo-osmotic shock.

*Bonny et al., 2021*, showed that *sic1* mutants (*sic1Δ* and *sic1(T173A)*) could adapt faster than wild-type cells to a hyperosmotic shock at the expense of increased cell death under repeated osmotic stresses. Consistent with their finding, we observed higher death rate of *sic1(T173A)* and *sic1(T173E)* mutant cells during repeated exposure to 1 M sorbitol at a period of 24 min in constant 2% glucose (*Figure 5—figure supplement 2a*). Surprisingly, we did not observe an increased death rate of these mutants under AS and IPS conditions (*Figure 5—figure supplement 2b*), when both hyperosmotic stress and glucose availability fluctuated periodically over time. In fact, under AS condition, the death rate of *sic1(T173A)* cells was even lower than the death rate of wild-type cells. Under this condition, the particularly low division rate of *sic1(T173A)* cells may lead to strengthened cell wall, decreasing the probability of cell bursting after hypo-osmotic shocks.

## HOG pathway mutants are fitter than wild-type cells under fast AS

The rates of cell division and cell death both contribute to fitness (i.e. the adaptive value) of a genotype in a particular environment. Since HOG pathway mutants exhibited different cell division and death rates compared to the wild-type genotype under AS, we calculated the population growth rate (division rate minus death rate) as a fitness estimate. Under IPS, the population growth rates of most mutant strains and of the wild-type strain were not significantly different; the only exception being the slightly lower growth rate of the *ste11Δ* mutant (*Figure 5d*). However, under AS with a fluctuation period of 24 min, several mutants had higher population growth rates than the wild-type strain (*Figure 5d*). In fact, the population growth rate was negative for the wild-type strain ($-1.4 \times 10^{-3}$ min$^{-1}$) as cells died faster than they divided and positive for the *hog1Δ* mutant ($1.0 \times 10^{-3}$ min$^{-1}$) and *pbs2Δ*

mutant ($5.5 \times 10^{-4}$ min$^{-1}$). These differences are clear in the microscopy images, as the population of wild-type cells visually shrank over time under AS (*Figure 5a*), while the population of *pbs2Δ* cells clearly expanded (*Figure 5b*). Thus, the *hog1Δ* and *pbs2Δ* genotypes are better adapted and would quickly outcompete wild-type cells under these dynamic conditions. However, this is only true when the frequency of environmental fluctuations is sufficiently high, since we did not observe significant differences in the population growth rate between the wild-type and *pbs2Δ* mutant under AS when the fluctuation period was 96 min (*Figure 5—figure supplement 1b*). We conclude that mutants that were first characterized by an inability to adapt to prolonged hyperosmotic stress can be well adapted when hyperosmotic stress rapidly fluctuates in antiphase with glucose availability. Therefore, the genetic mechanisms that contribute to adaptation under steady-state conditions could be detrimental under dynamic conditions, highlighting the importance of investigating how organisms adapt to dynamically changing environments.

## Osmoregulation is impaired under IPS but not under AS

The ability of cells to sense environmental fluctuations and to execute an adaptive response has been mainly studied using fluctuations of one stress cue at a time. How cells sense and respond to dual fluctuations of two interacting stresses remains a fundamental open question to understand how cells cope with complex environmental dynamics. Our findings suggest that the cell response to dual stress fluctuations can be very different depending on the phasing of the two stresses. Indeed, yeast cells appear to accumulate more glycerol under AS than under IPS. This could be either because of an impaired ability of cells to sense hyperosmotic shocks in absence of glucose or because of an impaired capacity to respond to hyperosmotic shocks in absence of glucose. Glycerol synthesis is regulated by the HOG pathway; thus, we investigated whether the activity of this pathway differed under IPS and AS conditions.

In the presence of glucose, activation of the HOG pathway in response to hyperosmotic shock triggers phosphorylation and nuclear translocation of Hog1p, which regulates transcription of multiple genes. A negative feedback loop dephosphorylates Hog1p, which leaves the nucleus in less than 15 min even if hyperosmotic stress is maintained (*Hersen et al., 2008*; *Muzzey et al., 2009*). To track HOG pathway activity, we quantified the nuclear enrichment of Hog1p over time by monitoring the subcellular location of Hog1 protein fused to a fluorescent marker (Hog1-GFP) in cells that also expressed the nuclear marker Htb2-mCherry. We did not detect any enrichment of Hog1-GFP in nuclei when only glucose fluctuated over time (*Figure 6—figure supplement 1a*), suggesting cells did not perceive glucose fluctuations as a significant osmotic cue. Although previous studies observed small transient (<2 min) peaks of Hog1-GFP nuclear localization after glucose was added back to the medium following glucose depletion (*Sharifian et al., 2015*; *Piao et al., 2012*), the temporal resolution in our experiments (one image every 6 min) may have been too low to detect these peaks.

In contrast, enrichment of Hog1-GFP fluorescence in nuclei was observed within minutes after exposure to 1 M sorbitol under both AS (*Figure 6a*) and IPS (*Figure 6b*). Therefore, cells can sense hyperosmotic shock, activate the HOG MAPK cascade, and phosphorylate Hog1 MAP kinase both in the presence (AS) and absence (IPS) of glucose. However, the adaptation dynamics of Hog1p (i.e. its exit from the nucleus) were remarkably different (*Figure 6a–c*): under AS, nuclear enrichment of Hog1-GFP peaked at 6 min following hyperosmotic shock and then quickly decayed and became undetectable after 30 min (*Figure 6a and c*)—essentially the same dynamics observed under periodic fluctuations of osmotic stress without glucose fluctuations (*Figure 6—figure supplement 1a*). Under IPS conditions, nuclear enrichment also peaked 6 min after hyperosmotic shock, but Hog1-GFP returned to the cytosol much more slowly; strong nuclear enrichment was still observed 48 min after exposure to hyperosmotic stress in the absence of glucose (*Figure 6b and c*). When the hyperosmotic stress was released, Hog1-GFP returned to the cytosol in less than 12 min.

To determine whether Hog1-GFP eventually returns to the cytoplasm during hyperosmotic stress in the absence of glucose, we applied a single pulse of 1 M sorbitol without glucose for 4 hr. Nuclear enrichment of Hog1-GFP reached basal levels about 2 hr after the onset of hyperosmotic stress (*Figure 6—figure supplement 1b*). The delayed exit of Hog1-GFP out of the nucleus under IPS suggests that the activity of the feedback loop regulating Hog1p dephosphorylation is impaired in the absence of glucose. We hypothesized that delayed nuclear export of Hog1-GFP under IPS could be due to impaired osmoregulation. In support of this hypothesis, we observed no recovery of cell size

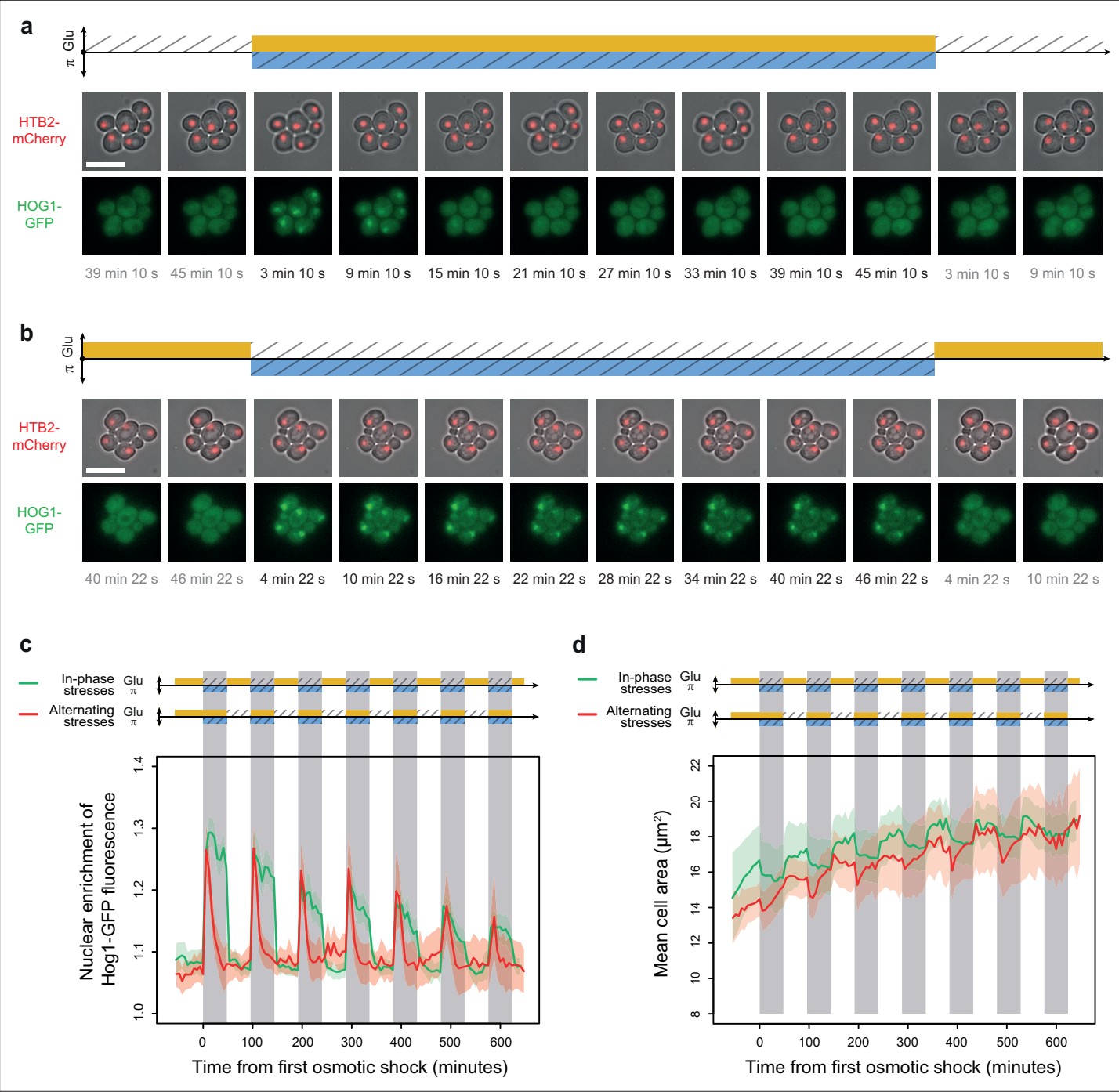

**Figure 6.** Osmoregulation is delayed after hyperosmotic stress under in-phase stresses (IPS) but not under alternating stresses (AS). (**a, b**) Time-lapse images of cells expressing Htb2-mCherry and Hog1-GFP under AS (**a**) and IPS conditions (**b**) for periods of 96 min, showing cellular localization of Hog1p during the second osmotic shock in each experiment. Top: fluorescence and bright-field images merged to visualize cell nuclei tagged with histone HTB2-mCherry. Bottom: fluorescence images showing Hog1-GFP localization. The time since the last environmental change is indicated below each image. Scale bars represent 10 μm. (**c**) Temporal dynamics of the enrichment of Hog1-GFP fluorescence in cell nuclei under IPS (red curve) and AS (green curve) conditions. Colored areas indicate 95% confidence intervals. (**d**) Temporal dynamics of cell size (area) in IPS (green) and AS (red) conditions for the same cells as in panel (**c**). Each curve shows the mean area measured among cells. Colored areas indicate 95% confidence intervals of the mean. (**c, d**) Each curve shows the mean nuclear enrichment or mean cell size for 11–25 cells in one or two fields of view. The colored graphs represent the periodic fluctuations of glucose (orange) and/or sorbitol (blue); hatching represents stress; gray indicates exposure to 1 M sorbitol. (**a–d**) Cells were grown under fluctuations of 2% glucose and 1 M sorbitol at a period of 96 min.

The online version of this article includes the following source data and figure supplement(s) for figure 6:

*Figure 6 continued on next page*

*Figure 6 continued*

**Source data 1.** Source data for *Figure 6*.

**Figure supplement 1.** Temporal dynamics of Hog1-GFP location under different regimes of environmental fluctuations.

**Figure supplement 1—source data 1.** Source data for *Figure 6—figure supplement 1*.

during hyperosmotic stress under IPS conditions (*Figure 6d*). Cells only returned to their initial size when sorbitol was removed and glucose was added back to the medium, which also corresponded to the moment when Hog1-GFP returned to the cytosol. In contrast, under AS, both the recovery of cell size and nuclear export of Hog1-GFP occurred while cells were still exposed to hyperosmotic stress (*Figure 6*), showing that osmoregulation was not impaired under these conditions. These results suggest glucose is necessary for the rapid osmoregulation that usually occurs in the first 20 min following hyperosmotic stress. This fast osmoregulation has been proposed to rely on the induction of glycerol synthesis via Hog1p-dependent post-translational mechanisms (*Schaber et al., 2012*). Since glucose is a metabolic precursor of glycerol, the absence of glucose may prevent glycerol synthesis and thereby fast osmoregulation. Further work will be necessary to test this hypothesis and study how glucose stored in the cell is used (or not) for glycerol production.

## Transcriptional response is impacted by the interaction between two environmental dynamics

Temporal fluctuations of two different stresses may have different or even opposite effects on gene expression, raising the question of how dual fluctuations of these two stresses would affect gene expression. Fast periodic fluctuation of osmotic stress was previously shown to cause hyper-activation of the *STL1* promoter ($P_{STL1}$) regulated by Hog1p (*Mitchell et al., 2015*), while glucose depletion is known to inhibit transcription (*Jona et al., 2000*) and translation initiation (*Ashe et al., 2000*; *Janapala et al., 2019*). We therefore asked whether dual fluctuations of glucose depletion and osmotic stress had additive effects on $P_{STL1}$ expression or whether one of the dynamic cues had a dominant effect. To address this question, we quantified the expression dynamics of a $P_{STL1}$-*mCitrine* fluorescent reporter gene under IPS and AS conditions. As expected, we observed transient expression of $P_{STL1}$-*mCitrine* in response to both short (48 min) and long (10 hr) pulses of 1 M sorbitol (*Figure 7*): the mean fluorescence level peaked at a 2.8-fold change 115 min after exposure to a short sorbitol pulse and 2.4-fold change 122 min after a long sorbitol pulse; then, fluorescence gradually returned to basal levels, even when osmotic stress was maintained. This expression pattern is consistent with previous quantifications of $P_{STL1}$ transcriptional activity during hyperosmotic stress (*Wosika and Pelet, 2020*; *Ben Meriem et al., 2019*). When hyperosmotic shocks were periodically applied for 48 min every 96 min, $P_{STL1}$-*mCitrine* expression constantly increased to reach 10.2-fold change after 11 hr (*Figure 7b*), suggesting a lack of adaptation to repeated osmotic shocks. Interestingly, hyper-activation of $P_{STL1}$-*mCitrine* transcription was also observed under AS conditions where $P_{STL1}$-*mCitrine* expression reached a maximum of 4.1-fold change after 413 min and remained as high as 3.1-fold change after 11 hr under the AS regime with a fluctuation period of 96 min (*Figure 7a and b*).

Conversely, we observed much weaker and slower induction of $P_{STL1}$-*mCitrine* expression under IPS: the maximal fold change was 0.7 after 11 hr of IPS with a fluctuation period of 96 min (*Figure 7a and b*). We observed a similar pattern (i.e. faster, stronger induction of $P_{STL1}$-*mCitrine* under AS than IPS) when the fluctuation period was 24 min (*Figure 7c*). Therefore, the *STL1* promoter is not activated by hyperosmotic stress in the absence of glucose, despite nuclear translocation of the MAP kinase Hog1p. This result suggests that the global repression of expression in response to abrupt glucose starvation is dominant over the hyper-activation of $P_{STL1}$ transcriptional activity induced by periodic osmotic stress.

## Discussion

Using microfluidics and time-lapse microscopy, we studied how yeast cells behave when confronted with two dynamic stresses that were applied either simultaneously (in-phase) or alternatively (in antiphase). This work demonstrates the value of applying artificial periodic fluctuations in several environmental parameters to understand how a biological system can process and integrate information from

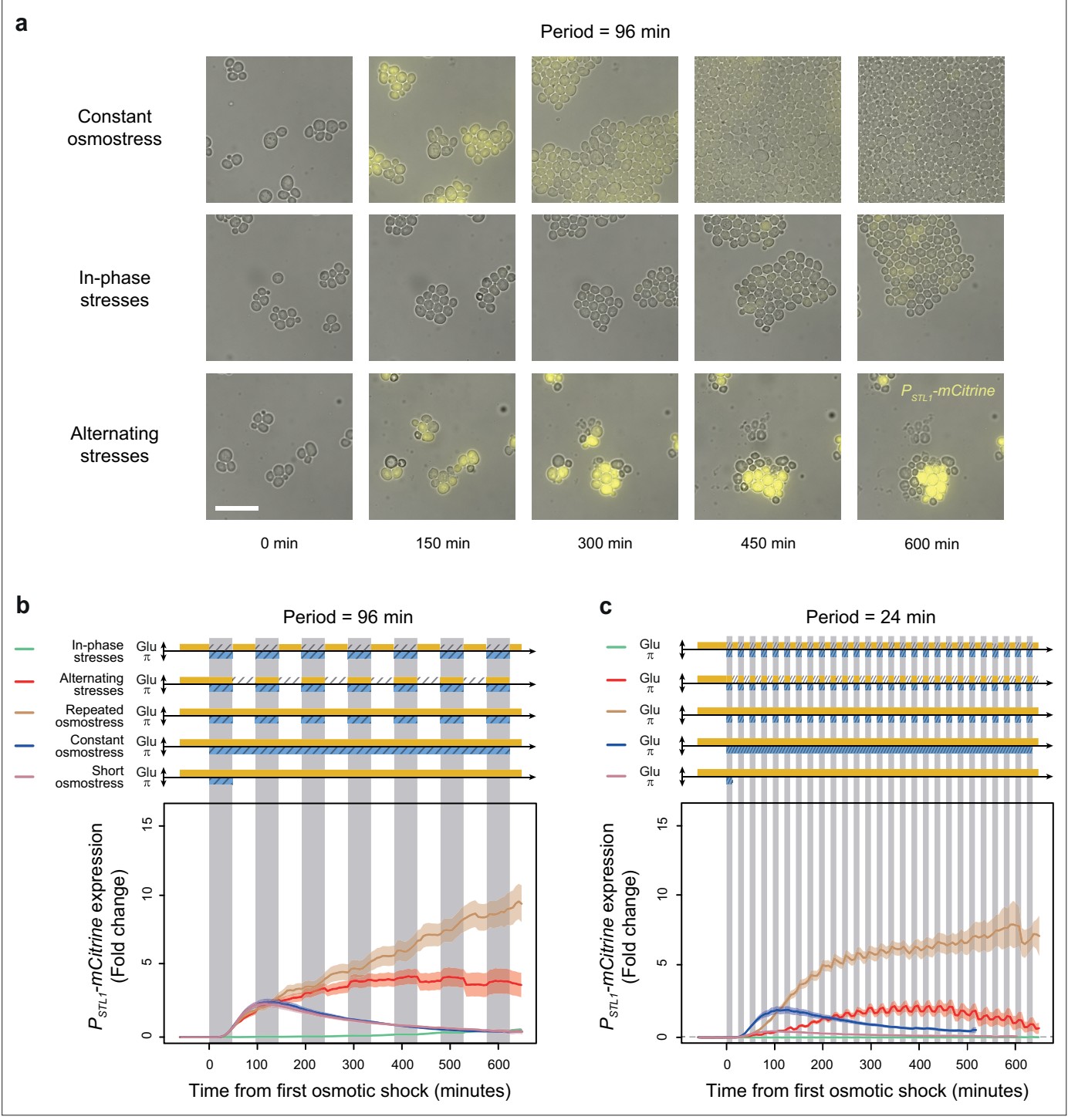

**Figure 7.** A transcriptional target of the HOG pathway is under-expressed during in-phase stresses (IPS) and over-expressed during alternating stresses (AS). (**a**) Images of cells expressing a fluorescent reporter (mCitrine) under control of the *STL1* promoter known to be regulated by the HOG pathway. Rows correspond to different conditions (as indicated on the left) and columns correspond to different time points after transitioning from complete medium to each condition. Bright-field and fluorescence images are overlaid. Scale bar represents 20 µm. (**b, c**) Temporal dynamics of $P_{STL1}$-*mCitrine* expression in five conditions: IPS (green curves), AS (red curve), periodic osmotic stress with constant glucose (brown curve), a single transition to constant osmotic stress with glucose for 10 hr (blue curve), and a short pulse of osmotic stress with constant glucose (purple curve). The fluctuation period is 96 min in (**b**) and 24 min in (**c**) for the first three conditions. 2% Glucose was used in all conditions. Each curve shows the mean fold change of fluorescence intensity measured for 30–65 cells from four or five fields of view. Colored areas indicate 95% confidence intervals of the mean. The colored graphs represent the periodic fluctuations of glucose (orange) and/or sorbitol (blue); hatching represents stress; gray indicates exposure to 1 M sorbitol.

*Figure 7 continued on next page*

*Figure 7 continued*

The online version of this article includes the following source data for figure 7:

**Source data 1.** Source data for *Figure 7*.

multiple cues in its environment. By using periodic inputs, we were able to investigate how changing the phasing of two stresses impacted fitness while keeping the duration of each stress constant. We found that dual fluctuations of glucose deprivation and osmotic stress had synergistic (non-additive) effects on cell proliferation that could not be easily predicted from the effects of fluctuating each stress separately. The phasing of the two stresses had a striking impact on several cell phenotypes, including division rate, death rate, osmoregulation, and transcriptional activation of a gene regulated by osmotic stress. The quantitative measurements of fitness that we collected for diverse genotypes under dynamic conditions with different glucose (main carbon source) concentrations can be used to further constrain mathematical models of yeast stress responses by adding two important features: resource allocation and regulation based on the presence of glucose. Moreover, our results indicate that the classic picture of yeast adaptation to osmotic stress cannot fully explain the behavior of yeast cells under fluctuating conditions. Indeed, by making glucose available for only short durations, we produced an environment in which the dynamics of the osmotic signaling pathway and its interaction with the glucose sensing pathway and glycolysis are critical. While the HOG signaling pathway was activated under all conditions, transcription of the target genes and inactivation of the HOG pathway were only detected when glucose and hyperosmotic stress were applied simultaneously, but not when hyperosmotic stress occurred in the absence of glucose. Therefore, if glucose is not available in the environment, cells are unable to commit to classical transcriptional/translational responses after osmotic stress. This finding may seem inconsistent with conclusions of a recent study showing that yeast cells displayed a stronger response to hyperosmotic stress at lower glucose concentration, including higher expression of several genes regulated by the HOG pathway (*Shen et al., 2023*). However, key differences in experimental procedures may explain this apparent discrepancy: *Shen et al., 2023*, investigated the response to a single hyperosmotic shock at diverse constant glucose concentrations—not dynamic stresses as we did—and the lowest glucose concentration they considered was 0.02% allowing slow cell growth—not complete glucose depletion that stopped growth. This further suggests that subtle changes in the metabolic environment of cells may have profound impact on their stress-response capacity.

Our findings shed lights on the importance of including different types of metabolic environment fluctuations when describing stress responses and probing gene regulatory networks with time-varying signals to constrain both mathematical and biological models of stress responses. Interestingly, another recent study showed that the interplay between osmotic stress and glucose concentration could cause bimodal expression of a key determinant of cell survival after starvation (*Lukačišin et al., 2022*). Since hyperosmotic stress was induced using high glucose concentration in this previous study, it would be particularly interesting to determine whether periodic fluctuations of hyperosmotic glucose concentration caused similar fitness defects as we observed when periodic fluctuations of glucose and sorbitol were applied in phase.

Importantly, in contrast to the well-known proliferative defects of osmo-sensitive mutants under constant hyperosmotic conditions, we show that periodically varying osmotic stress is not always detrimental to the growth of osmo-sensitive yeast cells. Indeed, HOG pathway mutants can grow and are even fitter than wild-type cells under fast alternating fluctuations of glucose deprivation and hyperosmotic stress. In this condition, osmo-sensitive mutant cells survive better than wild-type cells to repeated hypo-osmotic shocks, probably because they do not accumulate glycerol in response to hyperosmotic shocks and thus are less sensitive to fluctuations in the osmolarity of their environment. This also suggests that cells can be killed by exploiting their adaptive response. Forcing cells to repeatedly express their osmoadaptation program makes them very sensitive to osmotic rupture of the cell wall. Hence, periodically stressing cells (and their metabolic state) may be an efficient strategy to kill yeast cells. We further imagine that varying the timescale of fluctuations may even prevent cells from finding an evolutionary escape route.

Overall, we anticipate the importance of extending our study of the interaction of metabolic resources with other critical stresses, including antibiotic or mechanical stress, to higher eukaryotes. We propose that sending periodically a metabolic resource combined or alternating with a stressing

**Table 1.** Genotype of yeast strains used in this study.

| Name | Genotype | Mating type | Background | Reference or source |
|---|---|---|---|---|
| yPH_132 | *HTB2::HTB2-mCherry_pTEF-KanMX4-tTEF his3Δ1 leu2Δ0 met15Δ0 ura3Δ0* | a | BY4741 | This work |
| yPH_015 | *HTB2::mCherry-Ura3 HOG1-GFP* | a | BY4741 | Our team |
| yPH_091 | *pSTL1::yECITRINE-HIS5 his3Δ1 leu2Δ0 lys2Δ0 ura3Δ0 Hog1::mCherry-hph* | alpha | BY4742 | *Uhlendorf et al., 2012* https://doi.org/10.1073/pnas.1206810109 |
| yPH_051 | *pGPD1::YFP* | a | BY4741 | Gift from Megan McClean |
| yPH_403 | *HTB2::HTB2-mCherry_pTEF-KanMX4-tTEF pbs2Δ his3Δ1 leu2Δ0 met15Δ0 ura3Δ0* | a | yPH_132 | This work |
| yPH_405 | *HTB2::HTB2-mCherry_pTEF-KanMX4-tTEF fps1Δ his3Δ1 leu2Δ0 met15Δ0 ura3Δ0* | a | yPH_132 | This work |
| yPH_412 | *HTB2::HTB2-mCherry_pTEF-KanMX4-tTEF gpd1Δ his3Δ1 leu2Δ0 met15Δ0 ura3Δ0* | a | yPH_132 | This work |
| yPH_414 | *HTB2::HTB2-mCherry_pTEF-KanMX4-tTEF gpd2Δ his3Δ1 leu2Δ0 met15Δ0 ura3Δ0* | a | yPH_132 | This work |
| yPH_421 | *HTB2::HTB2-mCherry_pTEF-KanMX4-tTEF gpd1Δ gpd2Δ his3Δ1 leu2Δ0 met15Δ0 ura3Δ0* | a | yPH_132 | This work |
| yPH_441 | *HTB2::HTB2-mCherry_pTEF-KanMX4-tTEF hog1Δ his3Δ1 leu2Δ0 met15Δ0 ura3Δ0* | a | yPH_132 | This work |
| yPH_445 | *HTB2::HTB2-mCherry_pTEF-KanMX4-tTEF ste11Δ his3Δ1 leu2Δ0 met15Δ0 ura3Δ0* | a | yPH_132 | This work |
| yPH_452 | *HTB2::HTB2-mCherry_pTEF-KanMX4-tTEF glc3Δ his3Δ1 leu2Δ0 met15Δ0 ura3Δ0* | a | yPH_132 | This work |
| yPH_482 | *HTB2::HTB2-mCherry_pTEF-KanMX4-tTEF sic1(T173A) his3Δ1 leu2Δ0 met15Δ0 ura3Δ0* | a | yPH_132 | This work |
| yPH_486 | *HTB2::HTB2-mCherry_pTEF-KanMX4-tTEF sic1(T173E) his3Δ1 leu2Δ0 met15Δ0 ura3Δ0* | a | yPH_132 | This work |

agent allows to study the interplay between stress response and cell metabolism. We anticipate that such study could open novel research areas by revisiting the questions of stress response dynamics, within a system's view of cells homeostasis, in which metabolic activity regulates or interferes with multiple important cellular processes.

# Materials and methods

All materials produced in this study are available upon request.

## Yeast strains

All *S. cerevisiae* strains used in this study are derived from BY4741 or BY4742 (*Brachmann et al., 1998*) and are listed in *Tables 1 and 2*. The reference strain yPH_132 expresses the mCherry fluorescent reporter fused to histone H2B to label cell nuclei. To obtain this strain, the mCherry_pAgTEF-KanMX4-tAgTEF linear DNA fragment was amplified from plasmid pYM35 (PCR_TOOLBOX collection from EUROSCARF). This DNA fragment was inserted at the *HTB2* locus in strain BY4741 using a classic LiAc transformation protocol (*Gietz and Woods, 2002*) and a clone resistant to G418 was stored at −80°C as strain yPH_132.

All gene-deletion strains of yeast used in this study were obtained by directed mutagenesis of the yPH_132 strain (*HTB2::HTB2-mCherry*). Gene deletions were performed using CRISPR/Cas9 gene

**Table 2.** Oligonucleotides used to generate gene deletions and point mutations.

| Strain | Primer name | Sequence |
|---|---|---|
| | sgRNA_PBS2_F | GATCAATCAAAAGCGAGCAAGACAAGTTTTAGAGCTAG |
| | sgRNA_PBS2_R | CTAGCTCTAAAACTTGTCTTGCTCGCTTTGATT |
| | Deletion_PBS2_F | CGTCATACAACTAAAACTGATAAAGTACCCGTTTTCCGTACATTCTATAGATACATTATTATTAAGCAGATCGAGACGTTAATTTC |
| | Deletion_PBS2_R | GTAGCTTTTCGTCTGCTTTTTTTTGTTGTTATATTCACGTGCCTGTTTGCTTTTATTTGGATATTAACGGAAATTAACGTCTCGATCTG |
| pbs2Δ (yPH_403) | Seq_PBS2_F | GCTTACCTGCTTGCCGGAAG |
| | Seq_PBS2_R | CTATAACGAGTATAATGCAAG |
| | sgRNA_GPD1_F | GATCTCTGCTGCCATCCAAAGAGTGTTTTAGAGCTAG |
| | sgRNA_GPD1_R | CTAGCTCTAAAACACTCTTTGGATGGCAGCAGA |
| | Deletion_GPD1_F | TATACTACCATGAGTGAAACTGTTACGTTACCTTAAATTCTTTCTCCCTTTAATTTCTTTTATCTTACTCTCCTACATAAGACATCAAG |
| | Deletion_GPD1_R | ATGAAATATGATATAGAAGAGCCTCGAAAAAAGTGGGGGAAAGTATGATGTTATCTTTCTCCAATAAATCTTGATGTCTTATGTAGGAG |
| gpd1Δ (yPH_412) (yPH_421) | Seq_GPD1_F | GCACAACAAGTATCAGAATG |
| | Seq_GPD1_R | ATGCGGAAGAGGGTGTACAGC |
| | sgRNA_GPD2_F | GATCGCATTGGTCCGAAACCACCGGTTTTAGAGCTAG |
| | sgRNA_GPD2_R | CTAGCTCTAAAACCGGTGGTTTCGGACCAATGC |
| | Deletion_GPD2_F | TTTTTTTTATATATTAATTAATTTTATGTATTTGGTAGATTCAATTCTCTTTCCCTTTCCTTTCCTTCGCTCCCCTTCCTTATC |
| | Deletion_GPD2_R | ATAATGATAAATTGGTTGGGGAAAAGAGGCAACAGGAAAGATCAGAGGGGGGAGAGTGTGATAAGGAAGGGAGGGAGCGAAG |
| gpd2Δ (yPH_414) (yPH_421) | Seq_GPD2_F | CAGCTCTTCTCTACCCTGTC |
| | Seq_GPD2_R | GGTGATGTGATATGTAAACG |
| | sgRNA_FPS1_F | GATCACAGCAGGACAATTTCAACGGTTTTAGAGCTAG |
| | sgRNA_FPS1_R | CTAGCTCTAAAACCGTTGAAATTGTCCTGCTGT |
| | Deletion_FPS1_F | ATCAACAAAGTATAAACGCCTATTGTCCCAATAAAGCGTCGGTTGTTCTTCTTTATTATTTACCAAGTACGCTCGAGGGTACATTCTAATG |
| | Deletion_FPS1_R | TACCGGCGGGTAGTAAGCAGTATTTTTTTCTATCAGTCTATATTATTTCTTGTCTGTTTTCCATTAGAATGTACCCTCGAG |
| fps1Δ (yPH_405) | Seq_FPS1_F | CAGTGTGAATCCGGAGACGG |
| | Seq_FPS1_R | TACTTAAGACGATGGGTCAG |
| | sgRNA_HOG1_F | GATCGGCTCCTTACCACGATCCAAGTTTTAGAGCTAG |
| | sgRNA_HOG1_R | CTAGCTCTAAAACTTGGATCGTGGTAAGGAGCC |
| | Deletion_HOG1_F | TGGTAAATACTAGACTGCGAAAAAAGGAACAAAAGGGAAAACTACAACTATCGTATATAAAGTCCCTAACCACTCATTCTT |
| | Deletion_HOG1_R | TTCCTCTATACAACTATATACGTAAATACTTTTATGAGTACCATAAAAAAGAAGTAAGAAATGAGTGGTTAGGGAC |
| hog1Δ (yPH_441) | Seq_HOG1_F | TAGTGGAAGAGGAATTTGCG |
| | Seq_HOG1_R | GCCATAAGTGACGGTTCTTG |

*Table 2 continued on next page*

*Table 2 continued*

| Strain | Primer name | Sequence |
|---|---|---|
| | sgRNA_STE11_F | GATCTATGGTGCTTCTCAAGAAGGGGTTTTAGAGCTAG |
| | sgRNA_STE11_R | CTAGCTCTAAAACCCTTCTTGAGAAGCACCATA |
| | Deletion_STE11_F | CAGTAGAAAATATTCATATTACACACATGCATAAAGAGAGACCACTTAATAAAGCTAGTATGATAAAGATCACCGGTAGACGAAATATAC |
| | Deletion_STE11_R | ATGTATTATTTGATAAAAAATCGGCCAGAGCACTTAGTGCCATAAAAAAGAATTAATAAGTAGCCCTTTTGTATATTCGTCTACCGGTG |
| ste11Δ (yPH_445) | Seq_STE11_F | TTCTTTATGCTGTCCTCACC |
| | Seq_STE11_R | GAGAATCAAATACCGTCATC |
| | sgRNA_GLC3_F | GATCTTTCGACTACAGATTAGCAAGTTTTAGAGCTAG |
| | sgRNA_GLC3_R | CTAGCTCTAAAACTTGCTAATCTGTAGTCGAAA |
| | Deletion_GLC3_F | TCCTACATTTTTTTCCCTGATAACTTCCTGTTACTATTTAAGAACACCAAACCAAGTATAAAAGAACCGTCAAGAATAAAACTCTATACT |
| | Deletion_GLC3_R | GTACGTTTAGATATCTACCAATACATGAAGAGAAAAAAATTATTGAGTCTTGATTTTCAGTAAGCAATATAGTATAGAGTTTTATTCTTG |
| glc3Δ (yPH_452) | Seq_GLC3_F | TCGAGCCAAGTGACACCAGC |
| | Seq_GLC3_R | GACAGCTCTGCTATTCGCCC |
| | sgRNA_SIC1_F | GATCACCTGGTACGCCCAGCGACACAGTTTTAGAGCTAG |
| | sgRNA_SIC1_R | CTAGCTCTAAAACTGTCGCTGGGCGTACCAGGT |
| | Repair_SIC1_T173A_F | ACATTTATCACTTGAAAGAGATGAGTTTGATCAGACACATAGAAAGAAAGATTATTAAAGATGTACCTGGTGCGCCCAGCGACAAAGTGAT |
| | Repair_SIC1_T173A_R | TTCACTTTCTTGACTCCTGGCGTCATTTTTCGGAGAGTTGTTCCAATTCAAATGTTATCACTTTGTCGCTGGGCGC |
| | Repair_SIC1_T173E_F | ACATTTATCACTTGAAAGAGATGAGTTTGATCAGACACATAGAAAGAAAGATTATTAAAGATGTACCTGGTGAGCCCAGCGACAAAGTGAT |
| | Repair_SIC1_T173E_R | TTCACTTTCTTGACTCCTGGCGTCATTTTTCGGAGAGTTGTTCCAATTCAAATGTTATCACTTTGTCGCTGGGCTC |
| Sic1 (yPH_482 T173A) (yPH_486 T173E) | Seq_SIC1_F | CATTGGGTCGTGTAAATAGG |
| | Seq_SIC1_R | CTGAGTGACCAGTTCATCTG |

editing according to the method described in *Laughery et al., 2015*, that relies on the co-transformation of (i) a plasmid vector derived from pML104 (Addgene #67638) allowing expression of the Cas9 protein and a guide RNA specific to the target gene and (ii) a DNA repair fragment designed to delete the coding sequence of the target gene via homology-directed repair. For each gene deletion, a sequence containing the 20 nucleotides of the sgRNA recognition site in the coding sequence of the target gene was cloned between BclI and SwaI restriction sites in the pML104 plasmid by ligation of hybridized oligonucleotides. The DNA repair fragment was obtained by PCR amplification (Phusion Hot Start Flex 2X Master Mix, NEB M0536S) of two oligonucleotides with 20 bases of reverse complementarity to each other at the 3' end and 70 bases homologous to the region either immediately upstream or downstream of the coding sequence of the target gene at the 5' end. The repair fragment and the CRISPR/Cas9 plasmid specific to each gene were transformed together in exponentially growing yPH_132 cells using a standard lithium acetate method. Transformants were isolated on CSM-ura (Formedium, DCS0271) agar plates and gene deletions were confirmed by PCR screening and by Sanger sequencing. Positive clones were grown on YPG plates (10 g/L yeast extract, 20 g/L peptone, 5% [vol/vol] glycerol, 20 g/L bacto agar) to counter-select petite cells (p-phenotype) and then transferred on Complete Supplement Mixture (CSM) agar plates containing 0.8 g/L 5-fluoroorotic acid (Thermo Scientific R0812) to counter-select the CRISPR/Cas9 plasmid carrying the Ura3 gene. Two independent clones were stored at –80°C in 15% glycerol for each gene deletion (the clones used in this study are listed in *Table 1*).

## Microfluidics and live-cell imaging

Yeast cells were cultivated and imaged in custom-made microfluidic devices for all time-lapse microscopy experiments described in this study. One day before each experiment a new microfluidic chip was made by casting a mixture of 10 g of polydimethylsiloxane (Sylgard 184 kit, Neyco) and 1 g of curing agent on a master wafer made by soft lithography. The chip was then degased, cured at 65°C for 4 hr, peeled off, punched with a 1.2-mm-diameter needle at all positions of inlets and outlets and then bonded onto a 24×60 mm² coverslip after plasma activation of the surfaces. The chip pattern is shown in *Figure 1—figure supplement 1a*. In brief, it consists of five independent pairs of flow channels (800 µm wide×50 µm high) connected each to five growth chambers (400×400×3.8 µm³; L×W×H) where yeast cells are constrained to proliferate in monolayer.

Yeast strains were thawed from glycerol stocks kept at –80°C onto YPG agar plates (10 g/L yeast extract, 20 g/L peptone, 5% [vol/vol] glycerol, 20 g/L bacto agar) at least 3 days and no more than 2 weeks before each microscopy experiment. After 2 days of incubation at 30°C, YPG plates were kept at room temperature. The day before an experiment, a small amount of cells (~$10^5$ cells) was inoculated in 4 mL of CSM medium (6.7 g/L Yeast Nitrogen Base without amino acids [BD Difco], 2% glucose [Euromedex], 0.8 g/L CSM of amino acids [MP Biomedicals]) and incubated overnight at 30°C with 250 rpm orbital shaking. 100 µL of cell culture was then transferred to 5 mL of CSM medium and incubated for another 4 hr at 30°C to reach the exponential phase of growth. Cells were loaded in the microfluidic chip by injecting 50 µL of cell culture through each inlet of the chip using a small syringe.

Next, each pair of inlets was connected to a three-way solenoid valve (the Lee Company, LFAA1201418H) that was itself connected to two bottles of medium via tubings with 0.5 mm inner diameter and 1.5 mm outer diameter (Cole-Parmer Microbore Tubing). A custom-made valve controller piloted by a Node-RED application was used to dynamically control which medium was dispensed to the cells based on a predetermined schedule of valve state switching. Media were sterilized by filtration (0.22 µm) to remove large particles and their composition varied depending on the experiment. All media were based on 6.7 g/L Yeast Nitrogen Base without amino acids (BD Difco) and 0.8 g/L CSM of amino acids (MP Biomedicals) in ddH₂O. This base was complemented either with 2% glucose, 0.1% glucose, or no glucose and with 1 M sorbitol or no sorbitol. The outlets of the microfluidic chip were connected via tubings to a peristaltic pump (Ismatec IPC 12) and to an empty beaker to collect the flow-through.

The chip was then mounted on a motorized inverted microscope (Olympus IX83) equipped with LEDs for fluorescence excitation (CoolLED pE-300ultra), a Zyla 4.2 sCMOS camera (Andor—Oxford Instruments) and an autofocus module (IX3-ZDC2, Olympus). The microscope, the microfluidic system, and culture media were all placed inside an incubation chamber maintained at 30°C. Immediately after mounting the chip, the pump was turned on with a flow rate set at 120 µL/min for each outlet

tube, resulting in a flow rate of 240 μL/min in the growth chambers since they were each connected to two outlets. The medium dispensed to cells by default was CSM with either 2% glucose or 0.1% glucose. The microscope was controlled using iQ v3.6.3 software (Andor Technology) and all images were obtained using a 60× oil immersion objective (Olympus PlanApo N 60×/1.42) and a ×1.6 magnification changer. Using live bright-field imaging, we selected 25 positions (25 fields of view) of the motorized stage (Prior Scientific ProScan III) that captured 10–50 cells in each of the 25 growth chambers of the chip and were focused slightly below the median cell plane based on cell wall contrast. We used an iQ program that automatically scanned each position every 6 min for 12 or 24 hr and acquired bright-field and fluorescence images after autofocus adjustment. In parallel, we executed the program controlling the timing of valve state switches. This program always started with 1 hr of the default state corresponding to CSM+2% glucose or CSM+0.1% glucose so that cells could acclimate to the experimental settings and images recorded in the first hour were excluded from analyses. To detect YFP fluorescence, samples were exposed to blue LED illumination at an intensity of 10% using a 514/10 nm excitation filter and fluorescence was acquired with an exposure time of 250 ms using a 545/40 nm emission filter. To detect mCherry fluorescence, samples were exposed to green LED illumination at an intensity of 14% using a 560/40 nm excitation filter and fluorescence was acquired with an exposure time of 150 ms using a 630/75 nm emission filter. Microscopy images were saved in TIFF format and are available upon request.

Microscopy images were acquired following the same procedure for experiments aiming at quantifying division and death rates, for experiments aiming at quantifying the fluorescence of cells expressing $P_{STL1}$-*mCitrine* and for experiments aiming at quantifying the nuclear enrichment of Hog1-GFP fusion protein. However, image analysis was performed differently for each type of experiment as described below.

## Quantification of cell division and death rates

We used *ilastik* v1.3 for the segmentation and tracking of cell nuclei expressing HTB2-mCherry on fluorescence images. Three time-lapse movies of 120 fluorescence images of yPH_132 cells grown in CSM+2% glucose medium were used to train the machine learning algorithms to recognize the nuclei of single cells and the nuclei of dividing cells at various numbers of cells per image. The segmentation and tracking procedure was then applied to frames from all experiments after manually removing out-of-focus images that occasionally occurred due to autofocus failure. The output for each experiment was a CSV file for each of the 25 field of views where rows corresponded to all objects (i.e. cell nuclei) detected on all images and columns corresponded to parameters such as image identity, cell nucleus identity, parental nucleus identity, size of the nucleus, and *XY* coordinates of nucleus centroid. For all experiments involving combined stresses and AS (*Figures 3–6*), we manually added to CSV files a parameter indicating when cell death was observed based on visual inspection of bright-field images. This manual step was necessary because the nucleus of a dead cell could remain fluorescent and could be tracked for several hours after cell death. This time-consuming step was not performed in the analyses of other experiments where death events remained very rare (results shown in *Figure 2*). Next, we computed cell division rate and death rate using homemade R scripts. We designed and implemented an Eulerian measure of division rate that was robust to rare tracking errors and to cell saturation in the field of view. In this approach, we first define a tracking window of 1928×1928 pixels centered on each image of 2048×2048 pixels. We can then establish the relation:

$$N_{window_{(t+1)}} - N_{window_t} = N_{new_{(t)}} + N_{in_{(t)}} - N_{out_{(t)}}$$

where $N_{window_{(t)}}$ and $N_{window_{(t+1)}}$ are the number of cell nuclei in the tracking window at frames $t$ and $t+1$, respectively, $N_{new_{(t)}}$ is the number of divisions that occurred in the window between $t$ and $t+1$, $N_{in_{(t)}}$ is the number of cell nuclei that entered the window, and $N_{out_{(t)}}$ is the number of cell nuclei that exited the window between $t$ and $t+1$. From this relation, we computed $\frac{N_{new_{(t)}}}{N_{w_{(t)}}}$ which is the number of division events relative to the total number of cell nuclei in the window between $t$ and $t+1$. The slope of the linear regression of the cumulative sum $\sum_{t=t_0}^{t_1} \frac{N_{new_{(t)}}}{N_{window_{(t)}}}$ over time corresponded to the average division rate between $t_0$ and $t_1$. When death events were added to the input file, we only counted the nuclei of living cells when calculating the division rate. We excluded frames at the end of experiments that corresponded to an incomplete period of environmental fluctuation. The average death rate

was calculated using a similar approach: it corresponded to the slope of the linear regression of the cumulative sum $\sum_{t=t_0}^{t_1} \frac{N_{death_{(t)}}}{N_{window_{(t)}}}$ over time where $N_{death_{(t)}}$ was the number of death events observed in the tracking window between $t$ and $t+1$.

## Quantification of $P_{STL1}$-mCitrine expression

We used a custom image analysis approach based on the segmentation and tracking of cells in bright-field images to quantify $P_{STL1}$-mCitrine expression in single cells (yPH_091 strain) over time. First, a preprocessing step was necessary to eliminate out-of-focus images that occasionally occurred. This was done using a U-NET model trained to estimate the area of cells on each frame and to detect sudden changes of cell area between consecutive frames indicative of autofocus failure. Next, a proper cell segmentation was achieved via a machine learning algorithm based on the StarDist method (GitHub, https://stardist.net/) (*StarDist, 2021*; *Schmidt et al., 2018*; *Weigert et al., 2020*) version 0.7.3 that uses star-convex shape prior. This method was well suited to the round shape of yeast cells and performed slightly better than the U-NET network allowing more reliable tracking of cells. Training of StarDist and U-NET algorithms was performed using a GPU NVIDIA RTX Quadro. For each cell in a frame $F_t$, the tracking was performed by detecting which cell in the previous frame $F_{t-1}$ was closest to the cell on frame $F_t$ based on Euclidean distances between cell centroids. This simple method worked best for cells that did not move too much between consecutive frames: it could fail when the distance between two different cells in two consecutive frames was smaller than the distance between the same cell in the two frames. For this reason, we manually selected by visual inspection cells with correct tracking over at least 6 hr and excluded cells with wrong tracking in further analyses. We computed the mean fluorescence of each cell as the total fluorescence of the cell divided by the area of the cell. The area corresponded to the number of pixels classified as belonging to the cell in the segmented bright-field image. The total fluorescence of a cell was the sum of intensities of all pixels classified as belonging to the cell in the fluorescence image. Pixel classification of fluorescence images was performed using a *numpy.array* function in Python that applied the segmentation masks obtained from the analysis of bright-field images to the corresponding fluorescence images. R scripts were used to plot the mean fold change of fluorescence over time for all cells analyzed in images that were taken at the same time point in different growth chambers sharing the same regime of environmental fluctuations. Fold change was calculated for each cell as the difference of mean fluorescence observed for that cell in a given frame and the mean fluorescence observed among all cells in the first frame divided by the mean fluorescence among all cells in the first frame.

## Hog1-GFP nuclear enrichment

The nuclear enrichment of Hog1-GFP was quantified in cells expressing both the Hog1-GFP reporter and the nuclear marker Htb2-mCherry. Cell segmentation and tracking was performed on bright-field images following the same procedure as described above for the quantification of $P_{STL1}$-mCitrine expression. In addition, another segmentation was done for cell nuclei that were detected with the red fluorescence channel using a simple thresholding step. Each nucleus contour was then associated by contour overlapping comparison to its corresponding cell contour obtained by segmentation of bright-field images. We then computed the mean fluorescence in the green channel for each cell and for each nucleus. The mean fluorescence of a cell was calculated as the total intensity of all pixels classified as belonging to the cell (including the nucleus) divided by the number of these pixels. The mean fluorescence of a nucleus was calculated as the total intensity of all pixels classified as belonging to the nucleus divided by the number of these pixels. Finally, the nuclear enrichment of fluorescence was calculated for each cell as the mean fluorescence of the nucleus divided by the mean fluorescence of the cell. R scripts were used to plot the nuclear enrichment of Hog1-GFP fluorescence over time for all cells analyzed in images that were taken at the same time point in different growth chambers sharing the same regime of environmental fluctuations.

## Fluorescein assay

We used a fluorescein assay to characterize the temporal dynamics of medium fluctuations inside microfluidic chips. In this assay, we connected a microfluidic chip to CSM medium complemented with 50 nM fluorescein and to CSM medium without fluorescein. We programmed the valve to dispense CSM with fluorescein to the chip for 20 min followed by CSM without fluorescein for another 20 min and

repeated this treatment twice. We tried different flow rates on the peristaltic pump but only showed results for the optimal flow rate of 120 μL/min. We used the same microscopy setup as described above to image the growth chamber at the center of the chip, except that a 20× objective (Olympus Plan Achromat) was used to be able to visualize both the flow channel and the growth chamber in the field of view. One bright-field image and one fluorescence image were taken every 12 s. The fluorescence channel consisted of blue LED illumination at an intensity of 10% with a 514/10 nm excitation filter and acquisition with an exposure time of 250 ms using a 545/40 nm emission filter. We used ImageJ to quantify the mean fluorescence in two circular regions with a diameter of 280 pixels. One region was in the center of the growth chamber and the other region in the flow channel. A script in R was used to plot the relative level of fluorescence over time in each region. The relative fluorescence at time $t$ ($RF_t$) was calculated as $RF_t = \frac{F_t - F_{min}}{F_{max} - F_{min}}$, where $F_t$ is the mean fluorescence at time $t$, $F_{min}$ is the mean fluorescence observed between 30 and 40 min and between 70 and 80 min when fluorescein was at its minimal concentration in the chip and $F_{max}$ is the mean fluorescence observed between 10 and 20 min and between 50 and 60 min when fluorescein was at its maximal concentration in the chip.

## Acknowledgements

The authors would like to thank their team members for their critical reading of this manuscript. We also thank Williams Brett who helped us design the microfluidic control system. This work was supported by the European Research Council grant SmartCells (724813) and received support from grants ANR-11-LABX-0038, ANR-10-IDEX-0001-02, and ANR-16-CE12-0025-01.

## Additional information

### Funding

| Funder | Grant reference number | Author |
| --- | --- | --- |
| European Research Council | 724813 | Pascal Hersen |
| Agence Nationale de la Recherche | ANR-16-CE12-0025-01 | Pascal Hersen |
| Agence Nationale de la Recherche | ANR-10-IDEX-0001-02 | Pascal Hersen |
| Agence Nationale de la Recherche | ANR-11-LABX-0038 | Pascal Hersen |

The funders had no role in study design, data collection and interpretation, or the decision to submit the work for publication.

### Author contributions

Fabien Duveau, Conceptualization, Data curation, Software, Formal analysis, Investigation, Writing - original draft, Writing - review and editing; Céline Cordier, Data curation, Investigation, Writing - original draft, Writing - review and editing; Lionel Chiron, Software, Methodology, Writing - original draft; Matthias Le Bec, Resources, Investigation, Writing - original draft; Sylvain Pouzet, Resources, Data curation, Writing - original draft; Julie Séguin, Resources; Artémis Llamosi, Data curation, Formal analysis, Investigation; Benoit Sorre, Conceptualization, Formal analysis, Supervision, Writing - original draft, Writing - review and editing; Jean-Marc Di Meglio, Formal analysis, Supervision, Writing - original draft, Writing - review and editing; Pascal Hersen, Conceptualization, Supervision, Funding acquisition, Investigation, Writing - original draft, Writing - review and editing

### Author ORCIDs

Fabien Duveau http://orcid.org/0000-0003-4784-0640
Benoit Sorre http://orcid.org/0000-0001-8810-3298
Jean-Marc Di Meglio http://orcid.org/0000-0003-0446-4550
Pascal Hersen http://orcid.org/0000-0003-2379-4280

Reviewer #1 (Public Review): https://doi.org/10.7554/eLife.88750.3.sa1
Reviewer #2 (Public Review): https://doi.org/10.7554/eLife.88750.3.sa2
Author response https://doi.org/10.7554/eLife.88750.3.sa3

## Additional files

### Supplementary files
• MDAR checklist

### Data availability
All data analyzed in this study are included in the supporting files and are available on the following Zenodo archive: https://doi.org/10.5281/zenodo.10471016.

The following dataset was generated:

| Author(s) | Year | Dataset title | Dataset URL | Database and Identifier |
|---|---|---|---|---|
| Duveau F, Cordier C, Chiron L, LeBec M, Pouzet S, Séguin J, Llamosi A, Sorre B, Di Meglio J-M, Hersen P | 2024 | Lab513/Yeast cell responses and survival during periodic osmotic stress are controlled by glucose availability | https://zenodo.org/records/10471016 | Zenodo, 10.5281/zenodo.10471016 |

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
