## [Editor Report · eLife assessment]

This study presents **important** findings on how cells sense and respond to their surroundings, in particular when two environmental signals are presented periodically, in alternation or conjunction. The **compelling** analyses reveal some unexpected behaviors that could not have been drawn, from simpler experimental designs, related to the dynamic interplay between the starvation and hyper-osmotic stress responses in budding yeast, exemplifying that applying complex signals can unveil new biological insights, even for well-studied systems. The work will be of broad interest to researchers interested in fungal biology, dynamic systems, cell signaling, and cell biology.

---

## [Referee Report · Reviewer #1 (Public Review)]

In this study, the authors aimed to investigate how cells respond to dynamic combinations of two stresses compared to dynamic inputs of a single stress. They applied the two stresses - carbon stress and hyperosmotic stress - either in or out of phase, adding and removing glucose and sorbitol.

Both a strength and a weakness is that the cells' hyperosmotic response strongly requires glucose. For in-phase stress, cells are exposed to hyperosmotic shock without glucose, limiting their ability to respond with the well-studied HOG pathway; for anti-phase stress, cells do have glucose when hyperosmotically shocked, but experience a hypo-osmotic shock when both glucose and sorbitol are simultaneously removed. Responding with the HOG pathway and so amassing intracellular glycerol amplifies the impact of this hypo-osmotic shock. Counterintuitively then, it is the presence of glucose rather than the stress of its absence that is deleterious for the cells.

The bulk of the paper supports these conclusions with clean, compelling time-lapse microscopy, including extensive analysis of gene deletions in the HOG network and measurements of both division and death rates. The methodology the authors develop is powerful and widely applicable.

The authors' findings demonstrate the tight links that can exist between metabolism and the ability to respond to stress and the novel insights that can be gained using multiple dynamic inputs.

---

## [Referee Report · Reviewer #2 (Public Review)]

The authors have used microfluidic channels to study the response of budding yeast to variable environments. Namely, they tested the ability of the cells to divide when the medium was repeatedly switched between two different conditions at various frequencies. They first characterized the response to changes in glucose availability or in the presence of hyper-osmotic stress via the addition of sorbitol to the medium. Subsequently, the two stresses were combined by applying the alternatively or simultaneously (in-phase). Interestingly, they observed that the in-phase stress pattern allowed more divisions and low levels of cell mortality compared to the alternating stresses where cells were dividing slowly and many cells died. A number of mutants in the HOG pathway were tested in these conditions to evaluate their responses. Moreover, the activation of the MAPK Hog1 and the transcriptional induction of the hyper-osmotic stress promoter STL1 were quantified by fluorescence microscopy.

Overall, the manuscript is well structured and data are presented in a clear way. The time-lapse experiments were analyzed with high precision. The experiments confirm the importance of performing dynamic analysis of signal transduction pathways. While the experiments reveal some unexpected behavior, I find that the biological insights gained on this system remain relatively modest.

In the discussion section, the authors mention two important behaviors that their data unveil: resource allocation (between glycolysis and HOG-driven adaptation) and regulation of the HOG-pathway based on the presence of glucose. These types of behaviors had been already observed in other reports (Sharifan et al. 2015 or Shen et al. 2023, for instance). The experimental set-up used in this study provides highlights new aspects of the interplay between hyper-osmotic stress response and glucose availability.

The authors have tested various processes that could explain the slow growth observed in the alternating stress regime. Unfortunately, neither glycogen accumulation, cell-cycle arrest via Sic1 or the inhibition of protein production in starved cells could explain the observed behavior. However, one clear evidence that is presented is the link between glycerol accumulation during the sorbitol treatment and the cell death phenotype upon starvation in alternating stress condition.

One question which remains open is to what extent the findings presented here can be extended to other types of perturbations which for instance would combine Nitrogen limitation and hyper-osmotic stress.

---

## [Author Response]

The following is the authors’ response to the original reviews.

**Public Reviews:**

**Reviewer #1 (Public Review):**
In this study, the authors aimed to investigate how cells respond to dynamic combinations of two stresses compared to dynamic inputs of a single stress. They applied the two stresses - carbon stress and hyperosmotic stress - either in or out of phase, adding and removing glucose and sorbitol.Both a strength and a weakness, as well as the main discovery, is that the cells' hyperosmotic response strongly requires glucose. For in-phase stress, cells are exposed to hyperosmotic shock without glucose, limiting their ability to respond with the well-studied HOG pathway; for anti-phase stress, cells do have glucose when hyperosmotically shocked, but experience a hypo-osmotic shock when both glucose and sorbitol are simultaneously removed. Responding with the HOG pathway and so amassing intracellular glycerol amplifies the impact of this hypo-osmotic shock. Counterintuitively then, it is the presence of glucose rather than the stress of its absence that is deleterious for the cells.The bulk of the paper supports these conclusions with clean, compelling time-lapse microscopy, including extensive analysis of gene deletions in the HOG network and measurements of both division and death rates. The methodology the authors develop is powerful and widely applicable.Some discussion of the value of applying periodic inputs would be helpful. Cells are unlikely to have previously seen such inputs, and periodic stimuli may reveal behaviours that are rarely relevant to selection.

We thank the referee for his review. To answer the reviewer’s last comment, our main objective was not to study conditions that are ecologically relevant, but rather to perturb the system in an original way to reveal new mechanisms and properties of the system. The main advantage of periodic inputs over more complex or unpredictible types of temporal fluctuations is that they can be defined with few parameters that are easy to interpret and to integrate in biophysical models. For instance, by using periodic inputs we were able to investigate how changing the phasing of two stresses impacted fitness while keeping other parameters constant (the duration of each stress was kept constant). We added two sentences at the beginning of the discussion to highlight the value of using periodic inputs.

We do not fully agree with the reviewer’s statement that periodic stimuli may reveal behaviours that are rarely relevant to selection. Indeed, many parameters of natural environments are known to vary periodically, such as light, temperature, predation, tides. Even if the periodic stimuli we use are artificial, they can still be a valuable tool to reveal new molecular processes. For instance, null mutants have been invaluable to understand biological systems despite being unlikely to reveal behaviours relevant to selection.

The authors' findings demonstrate the tight links that can exist between metabolism and the ability to respond to stress. Their study appears to have parted somewhat from their original aim because of the HOG pathway's reliance on glucose. It would be interesting to see if the cells behaviour is simpler in periodically varying sorbitol and a stress where there is little known connection to the HOG network, such as nitrogen stress.

The use of periodic nitrogen stress is a very interesting suggestion from both reviewers. However, we think it represents a large amount of work that deserves its own study. In particular, it would require first identifying a relevant period at which nitrogen fluctuations have an impact on division rate similar to what we observed for glucose fluctuations before performing experiments in AS and IPS conditions.

Nitrogen starvation is known to induce filamentous growth via activation of components of the HOG pathway (Cullen and Sprague, 2012), with potential cross-talk between filamentous growth and hyperosmotic stress response. Therefore, periodic osmotic stress and periodic nitrogen starvation may interact in a complex way.

**Reviewer #2 (Public Review):**
The authors have used microfluidic channels to study the response of budding yeast to variable environments. Namely, they tested the ability of the cells to divide when the medium was repeatedly switched between two different conditions at various frequencies. They first characterized the response to changes in glucose availability or in the presence of hyper-osmotic stress via the addition of sorbitol to the medium. Subsequently, the two stresses were combined by applying the alternatively or simultaneously (in-phase). Interestingly, the observed that the in-phase stress pattern allowed more divisions and low levels of cell mortality compared to the alternating stresses where cells were dividing slowly and many cells died. A number mutants in the HOG pathway were tested in these conditions to evaluate their responses. Moreover, the activation of the MAPK Hog1 and the transcriptional induction of the hyper-osmotic stress promoter STL1 were quantified by fluorescence microscopy.Overall, the manuscript is well structured and data are presented in a clear way. The time-lapse experiments were analyzed with high precision. The experiments confirm the importance of performing dynamic analysis of signal transduction pathways. While the experiments reveal some unexpected behavior, I find that the biological insights gained on this system remain relatively modest.In the discussion section, the authors mention two important behaviors that their data unveil: resource allocation (between glycolysis and HOG-driven adaptation) and regulation of the HOG-pathway based on the presence of glucose. These behaviors had been already observed in other reports (Sharifan et al. 2015 or Shen et al. 2023, for instance). I find that this manuscript does not provide a lot of additional insights into these processes.

We thank the referee for his review. We agree with the reviewer that the interaction between glucose availability and osmotic stress response has been investigated in previous studies. However, this interaction was investigated using experimental procedures that differed from our approach in critical ways, and therefore the behaviors observed were not the same. In Sharifian et al. (2015), the authors identified a new negative feedback loop regulating Hog1 basal activity and described underlying molecular mechanisms. This feedback loop is unlikely to explain differences of cell fitness we observed in IPS and AS conditions, because (1) differences of division rate was still observed in hog1 mutant cells and (2) differences of death rate involve glycerol synthesis, which is independent of the feedback loop described in Sharifian et al. (2015). In Shen et al. (2023), the authors observed a stronger expression of Hog-responsive genes at lower glucose concentrations, which seems contradictory with our observation of very low pSTL1-GFP expression in absence of glucose. However, they did not use fluctuating conditions and they did not report expression of stress-response genes when glucose was totally depleted (the lower glucose concentration they used was 0.02%) as we did, which may explain the different outcomes. We added three sentences in the discussion to compare our findings to those of Shen et al. (2023).

One clear evidence that is presented, however, is the link between glycerol accumulation during the sorbitol treatment and the cell death phenotype upon starvation in alternating stress condition. However, no explanations or hypothesis are formulated to explain the mechanism of resource allocation between glycolysis and HOG response that could explain the poor growth in alternating stresses or the lack of adaptation of Hog1 activity in absence of glucose.

In the revised version of the manuscript, we included a new result section and a supplementary figure (Figure 4 – figure supplement 2) where we tested three hypotheses to explain the lower division rate observed in AS condition relative to IPS condition. We found no evidence supporting these hypotheses, and the mechanisms responsible for the reduced growth in AS condition therefore remains elusive.

Another key question is to what extent the findings presented here can be extended to other types of perturbations. Would the use of alternative C-source or nitrogen starvation change the observed behaviors in dynamic stresses? If other types of stresses are used, can we expect a similar growth pattern between alternating versus in-phase stresses?

As mentioned above in our response to the other reviewer, these are very interesting questions that we think go beyond the scope of our study due to the amount of work involved.

**Recommendations for the authors:**

**Reviewer #1**
My comments are only minor.More paragraphs would improve legibility.

To improve legibility, we split the longer section of the Results in three paragraphs page 12, section entitled “Osmoregulation is impaired under in-phase stresses but not under alternating stresses.” However, we kept it as one section with a single title for global coherency: each section of the results corresponds to one main figure and have one main conclusion.

I found AS and IPS confusing because what becomes important is whether sorbitol appears with glucose or not. For me, an acronym that makes that co-occurrence clear would be better or even better still no acronyms at all.

We tried several alternative names for the two conditions in previous drafts of the manuscript. Based on colleagues feedback, AS and IPS acronyms appeared as a good compromise between concision and clarity. To avoid confusion, the two acronyms are precisely defined when they are first used in the Results section. We think it is more important to emphasize the co-occurrence (or not) of the two stresses, rather than the co-occurrence of glucose and sorbitol. Indeed, standard yeast medium contains glucose but no sorbitol, and therefore we defined the two periodic conditions based on differences from standard medium. Even though we avoided using acronyms as much as possible in the manuscript, the use of these two acronyms to refer to the dual fluctuations of the environment seemed essential for concision. Indeed, IPS and AS acronyms are used many times in the results (16 occurrences on page 12 alone), figures and figure legends.

I would consider moving some of Fig S2 to the main text: it helps clarify where Fig 2 is coming from and is referenced multiple times.

We fully agree with the reviewer and we moved panels A-D from Figure S2 to the main Figure 2.

On page 10, "constantly facing a single stress that changes over time" is confusing. Perhaps "repetitively facing a single stress" instead?

We agree this sentence could be wrongly interpreted the way it was written. We changed it to: “cells grow more slowly when facing periodic alternation of the two stresses (AS) than when facing periodic co-occurrence of these stresses (IPS)”.

Is there any knowledge on how cells resist hyperosmotic stress in the absence of glucose? That would help explain the IPS results.

Based on comments from both reviewers, we surveyed the literature to flesh out the discussion of hypotheses that would help explain observed differences between AS and IPS conditions. We found few studies that investigated cell responses in the absence of glucose, and because of significant differences in the experimental approaches it remains difficult to explain our results from conclusions of these previous studies. For instance, Shen et al., 2023 described and modeled the hyperosmotic stress response at various glucose concentrations. They found that Hog1p relocation to the nucleus after hyperosmotic shock lasted longer at lower glucose concentration, which is consistent with our finding in absence of glucose. However, they did not include the absence of glucose in their experiments or periodic fluctuations of glucose concentration. In addition, their model ignores the impact of cell signaling processes involved in growth arrest in response to hyperosmotic stress or glucose depletion. It is therefore difficult to relate their conclusions to our results. We have developed the discussion of our study to include these hypotheses and to clarify what is explained or not in our IPS and AS results.

There is knowledge on activation of the hyperosmotic stress pathway in response to glucose fluctuations, but not about the response to hyperosmotic stress in absence of glucose.On page 11, Figure 5a should be Figure 4a.

Correct.

I would explain the components of the HOG pathway in the caption of Fig 1 or in the text when you cite Fig 1a. They are described later, but an early overview would be useful.

To give more context, we added the following sentences to the caption of Figure 1: “Yeast cells maintain osmotic equilibrium by regulating the intracellular concentration of glycerol. Glycerol synthesis is regulated by the activity of the HOG MAP kinase cascade that acts both in the cytoplasm (fast response) and on the transcription of target genes in the nucleus (long-term response). For simplicity, we only represented on the figure genes and proteins involved in this study.”

On page 16, I wasn't sure what "redirect metabolic fluxes against glycerol synthesis" meant.

For more clarity, we modified this sentence to: “Since glucose is a metabolic precursor of glycerol, the absence of glucose may prevent glycerol synthesis and thereby fast osmoregulation."

For Fig 2, having a dot-dash and dash-dash lines rather than both dash-dash would be better.

We made the proposed change, assuming the reviewer was referring to the gray dashed lines and not the colored ones.

In the caption of Fig 3, 2% glucose is 20 g/L.

We thank the reviewer for catching this typo.

In the Materials and Methods Summary, adding how you estimated death rates would be helpful: they are not often reported.

The calculation of death rates was explained in the Methods section. For more clarity, we modified the names of the parameters in the equation to make more explicit which ones refer to cell death.

**Reviewer #2 (Recommendations For The Authors):**
In Figure 2, it would be interesting to show individual growth rates of the perturbations at various frequencies as shown in Figures 3 c and d.

We thank the reviewer for this suggestion. We added a new supplementary figure (Figure 2 – figure supplement 2) showing the temporal dynamics of division rates at three different frequencies of osmostress and glucose depletion. We did not include high frequencies (periods below 48 minutes) because the temporal resolution of image acquisition in our experiments (1 image every 6 minutes) was too low. Very interestingly, this new analysis suggests that the positive relationship between the frequency of glucose depletion and division rate is explained by a delay between glucose removal and growth arrest rather than a delay between glucose addition and growth recovery. We therefore added the following conclusion:

“Under periodic fluctuations of 2% glucose, the division rate was lower during half-periods without glucose than during half-periods with glucose (Figure 2 – figure supplement 2d-f), as expected. However, this difference depended on the frequency of glucose fluctuations: the average division rate during half-periods without glucose was higher at high frequency (small period) than at low frequency (large period) of fluctuations (Figure 2 – figure supplement 2d-f). Therefore, the effect of the frequency of glucose availability on the division rate in 2% glucose is likely due to a delay between glucose removal and growth arrest: cell proliferation never stops when the frequency of glucose depletion is too fast.”

According to Sharifan et al. 2015, I would have expected that Hog1 would not relocate in the nucleus in 0% glucose. I wonder if this is due to the use of sorbitol as a stressor or the presence of low levels of glucose in the medium. I would suggest performing some control experiments with NaCl as hyperosmotic agent and test the addition of 2-deoxy-glucose to completely block glycolysis.

After careful reading of Sharifian et al. 2015, we fail to understand why the reviewer think Hog1 would be expected to not relocate to the nucleus after hyperosmotic stress in 0% glucose. In this previous study, the authors never combined glucose depletion with a strong hyperosmotic stress as we did in our study. They report the results of independent experiments where cells were exposed either to a single pulse of hyperosmotic stress (0.4 M NaCl) or to transient glucose starvation, but they did not combine these two stimuli. In this context, it is difficult to compare their results with ours. The fact that Sharifian et al. 2015 did not observe Hog1 nuclear relocation in 0% glucose (consistent with our result in Figure 6 – figure supplement 1a, yellow curve) is not inconsistent with our observation of Hog1 nuclear enrichment in 0% glucose + 1M sorbitol. One potential discrepancy between the two studies is the fact that they observed a small transient peak of Hog1 nuclear localization just after glucose is added back to the medium, while we failed to observe this peak in similar conditions (yellow curve in Figure 6 – figure supplement 1a). However, this could be simply explained by the temporal resolution of our experimental system: we image cells once every 6 minutes and the peak lasts less than 2 minutes in Sharifian et al. 2015. We added a sentence to discuss this minor point in the Results: “Although previous studies observed small transient (less than two minutes) peaks of Hog1-GFP nuclear localization after glucose was added back to the medium following glucose depletion (Sharifian et al., 2015, Piao et al., 2013), the temporal resolution in our experiments (one image every 6 minutes) may have been too low to detect these peaks.”.

While we agree many additional experiments would be interesting, such as testing the effects of different stress factors or the non-metabolizable glucose analog 2-deoxy-D-glucose, we think this is beyond the scope of this study because such experiments are likely to open broad perspectives and to not be conclusive in a reasonable amount of time.

When discussing Figure 7, the authors write that the HOG pathway is "overactivated" or "hyperactivated". I would refrain from using these terms because as seen in Figure 6, the Hog1 activity pattern, if anything, decreases as the number of alternative pulses increases. The high level of pSTL1mCitrine measured is mostly due to the long half-life of the fluorescent protein.

We used the formulation “hyper-activation” of the HOG pathway because Mitchell et al. 2015 used it to refer to the same phenomenon in their seminal study. This "hyper-activation" refers to the fact that both the integral activation of Hog1p (sum of areas under Hog1 nuclear peaks) and the global activation of transcriptional targets is much higher during fast periodic hyperosmotic stress than during constant hyperosmotic stress. That being said, we understand the point made by the reviewer about the decreasing size of Hog1 peaks over time during repeated pulses of osmotic stress. Therefore, we slightly modified the text to refer to hyper-activation of pSTL1-mCitrine transcription or expression instead of hyper-activation of the HOG pathway. For coherency, we replaced all instances of “overactivation” by “hyper-activation”.

Last but not least, the high level of pSTL1-mCitrine is both due to the long half-life of the protein and to the fact that pSTL1 transcription is never turned off due to high Hog1p activity under fast periodic osmostress.

Minor comments:In the main text, I think it might be more intuitive to refer to doubling time in hours instead of division rates in 1/min which are harder to interpret.

In an early draft of the manuscript, we made figures with either division rates or with doubling times (ln(2)/division rate) and we received mixed opinions from colleagues on what measure was more intuitive to interpret. Both measures are widely used in the literature, and we decided to use division rates in the final version of the figures because it was more directly related to population growth rate and to fitness. For instance, the population growth rate shown in Figure 5 is simply calculated by subtracting the death rate from the division rate. For coherency, we therefore reported division rates instead of doubling times in figures and results. However, to address the reviewer’s comment we included the doubling times (in addition to the division rates) when mentioning the most important results. For instance, page 12: “Strikingly, cells divided about twice as fast under IPS condition (1.67 x 10-3 division/min, corresponding to an average doubling time of 415 minutes) than under AS condition (9.4 x 10-4 division/min, corresponding to an average doubling time of 737 minutes)”.

I found various capitalized version of "HOG /Hog pathway"

We corrected this incoherency and used “HOG pathway” everywhere.

Page 11. Figure 5a should refer to Figure 4a I believe.

Correct.

The methods are generally very thorough and precise. The explanation about the calculation of the division rate seems incomplete. For completeness, it would be good to mention the brand and model of valves used. In addition, it would be interesting to have an idea of the number of cells and microcolonies tracked in the various growth experiments.

We are not sure why the reviewer found the explanation of the calculation of division rate incomplete. For more clarity, we modified the names of parameters in the equations to make them more explicit. We also added a reference to Supplementary File 1 that contains all R scripts used to calculate division rates and death rates. We included the brand and model of valves used, as requested. As for the number of cells tracked in the various experiments, we mentioned in the Methods: “we selected 25 positions (25 fields of view) of the motorized stage (Prior Scientific ProScan III) that captured 10 to 50 cells in each of the 25 growth chambers of the chip and were focused slightly below the median cell plane based on cell wall contrast.” To address the reviewer’s comment, we also included the range of number of tracked cells for each experiment in corresponding figure legends.